# Distinct pathogenic mutations in ARF1 allow dissection of its dual role in cGAS-STING signalling

Johannes Lang[1], Tim Bergner[2], Julia Zinngrebe[3], Alice Lepelley [iD][4], Katharina Vill[5], Steffen Leiz[6], Meinhard Wlaschek [iD][7], Matias Wagner[8,9], Karin Scharffetter-Kochanek[7], Pamela Fischer-Posovszky[3,10], Clarissa Read [iD][2], Yanick J Crow[4,11], Maximilian Hirschenberger [iD][1 ✉] & Konstantin M J Sparrer [iD][1,12 ✉]

## Abstract

**Tight control of cGAS-STING-mediated DNA sensing is crucial to avoid auto-inflammation. The GTPase ADP-ribosylation factor 1 (ARF1) is crucial to maintain cGAS-STING homeostasis and various pathogenic ARF1 variants are associated with type I interferonopathies. Functional ARF1 inhibits STING activity by maintaining mitochondrial integrity and facilitating COPI-mediated retrograde STING trafficking and deactivation. Yet the factors governing the two distinct functions of ARF1 remained unexplored. Here, we dissect ARF1's dual role by a comparative analysis of disease-associated ARF1 variants and their impact on STING signalling. We identify a de novo heterozygous s.55 C > T/p.R19C ARF1 variant in a patient with type I interferonopathy symptoms. The GTPase-deficient variant ARF1 R19C selectively disrupts COPI binding and retrograde transport of STING, thereby prolonging innate immune activation without affecting mitochondrial integrity. Treatment of patient fibroblasts in vitro with the STING signalling inhibitors H-151 and amlexanox reduces chronic interferon signalling. Summarizing, our data reveal the molecular basis of a ARF1-associated type I interferonopathy allowing dissection of the two roles of ARF1, and suggest that pharmacological targeting of STING may alleviate ARF1-associated auto-inflammation.**

**Keywords** ARF1; cGAS; Interferonopathy; Interferon; STING
**Subject Categories** Immunology; Membranes & Trafficking; Molecular Biology of Disease

## Introduction

The innate immune system is pivotal in defence against pathogens such as bacteria and viruses. Upon pathogen recognition, a signalling cascade is triggered, culminating in the synthesis of type I interferons (IFNs) and other (pro-)inflammatory cytokines (Cao, 2016; Li and Wu, 2021). The secreted IFNs engage the type I interferon α/β receptor (IFNAR), subsequently activating the Janus kinase/signal transducer and activator of transcription (JAK/STAT) pathway (Ivashkiv and Donlin, 2014). This results in the induction of a transcriptional programme upregulating interferon-stimulated genes (ISGs), many of them well-known anti-viral factors, thus establishing an antimicrobial state (Wang et al, 2017). While crucial to prevent and combat infections, chronic type I IFN signalling causes human pathology, as evidenced by the type I interferonopathies—monogenic diseases characterized by enhanced type I IFN signalling (Crow, 2011).

The cyclic GMP–AMP synthase (cGAS) (Sun et al, 2013; Gao et al, 2013)—stimulator of interferon genes (STING) (Ishikawa and Barber, 2008; Ishikawa et al, 2009) pathway is a well-studied prototypic example of an innate immune sensing pathway previously implicated in a number of type I interferonopathies. Recognition of double-stranded (ds)DNA by cGAS results in the synthesis of the second messenger 2′3′ cyclic GMP–AMP (cGAMP) (Ablasser et al, 2013; Wu et al, 2013). In the resting state, STING is sequestered at the endoplasmic reticulum (ER) through interaction with stromal interaction molecule 1 (STIM1) (Srikanth et al, 2019) and autoinhibition (Ergun et al, 2019; Liu et al, 2023). cGAMP binding induces structural rearrangements (Zhang et al, 2013) and STING oligomerization (Shang et al, 2019), prompting its translocation from the ER to the ER–Golgi intermediate compartment (ERGIC) and the Golgi via COPII vesicles (Sun et al, 2018; Gui et al, 2019). Palmitoylation of STING at the Golgi facilitates its clustering in lipid rafts at the trans-Golgi network (TGN) (Mukai et al, 2016). Subsequently, TANK-binding kinase 1 (TBK1) is

[1]Institute of Molecular Virology, Ulm University Medical Center, Ulm, Germany. [2]Central Facility for Electron Microscopy, Ulm University, Ulm, Germany. [3]Department of Pediatrics and Adolescent Medicine, Ulm University Medical Center, Ulm, Germany. [4]Institut Imagine-Inserm UMR1163, Laboratory of Neurogenetics and Neuroinflammation, Université Paris Cité, Paris, France. [5]Department of Pediatric Neurology and Developmental Medicine, Dr. von Hauner Children's Hospital, LMU-University of Munich, Munich, Germany. [6]Division of Neuropediatrics, Klinikum Dritter Orden, Munich, Germany. [7]Department of Dermatology and Allergic Diseases, Ulm University Medical Center, Ulm, Germany. [8]Institute of Human Genetics, Klinikum rechts der Isar, School of Medicine and Health, Technical University of Munich, Munich, Germany. [9]Institute of Neurogenomics, Helmholtz Zentrum Munich, Munich, Germany. [10]German Center for Child and Adolescent Health (DZKJ), Partner site Ulm, Ulm, Germany. [11]MRC Human Genetics Unit, Institute of Genetics and Cancer, University of Edinburgh, Edinburgh, UK. [12]German Center for Neurodegenerative Diseases (DZNE), Ulm, Germany. ✉E-mail: maximilian.hirschenberger@uni-ulm.de; Konstantin.Sparrer@uni-ulm.de

recruited, undergoes autoactivation, and then phosphorylates STING at serine 366 (Ogawa et al, 2018; Tanaka and Chen, 2012; Liu et al, 2015). Post phosphorylation, STING oligomerizes and engages with interferon regulatory factor 3 (IRF3), initiating the transcription of type I IFNs (Tanaka and Chen, 2012; Fitzgerald et al, 2003). To terminate this signalling, STING undergoes lysosomal degradation (Gonugunta et al, 2017; Gentili et al, 2023; Kuchitsu et al, 2023) and/or is transported back to the ER via COPI vesicle-mediated retrograde trafficking (Lepelley et al, 2020; Deng et al, 2020; Mukai et al, 2021; Steiner et al, 2022).

ADP-ribosylation factors (ARFs), particularly ARF1, play a pivotal role in intracellular trafficking by recruiting coat proteins, such as COPI components for vesicle budding from the Golgi (Volpicelli-Daley et al, 2005; Popoff et al, 2011; Letourneur et al, 1994). Recent research showed that ARF1 maintains cGAS-STING homeostasis by both promoting mitochondrial integrity and aiding in the retrograde transport of STING from the Golgi to the ER. De novo variants in *ARF1* have been reported to result in enhanced type I interferon signalling in patients (Hirschenberger et al, 2023). Of particular note, the substitution p.R99C identified in affected individuals impairs both functions of ARF1, inducing mitochondrial DNA leakage, which triggers cGAS activation, and hampering cGAS-STING signal termination, resulting in persistent IFN signalling and an autoinflammatory phenotype (Hirschenberger et al, 2023). Importantly, it remains unclear whether the two defined functions of ARF1 in cGAS-STING signalling have a distinct basis, or if they both result from a disturbance of the trafficking activity of ARF1.

In this study, we present data relating to IFN signalling due to a previously unstudied patient-associated variant in *ARF1* (NM_001024226.1) i.e. c.55 C > T, p.Arg19Cys (R19C). In contrast to R99C, R19C specifically perturbs the retrograde transport of STING without impacting mitochondrial integrity. Nevertheless, impaired retrograde transport was sufficient to induce aberrant IFN signalling after initial stimulation. Furthermore, through an in vitro proof-of-concept treatment approach, we demonstrate the potential to limit IFN signalling in primary patient cells. Our data indicates that the two roles of ARF1 can be separated. By dissecting the processes of retrograde transport and mitochondrial integrity, we gain a deeper understanding of ARF1's involvement in these regulatory mechanisms.

## Results and discussion

### Mutation R19C of ARF1 induces STING-dependent IFN signalling

In an ongoing monitoring of patients with undefined clinical phenotypes for type I IFN upregulation, we identified a male (AGS3238) with a de novo heterozygous c.55 C > T/p.R19C substitution in the *ARF1* gene. Neurodevelopmental examination revealed delays in motor skills and speech, accompanied by oral-motor coordination difficulties. An MRI scan showed periventricular nodular heterotopia (PVNH). Notably, no skin lesions were observed. At age 20 months an increased expression of interferon-stimulated genes (ISGs) was recorded in patient whole blood (IFN score of 7.615, cut-off: 2.758, Fig. 1A), consistent with a type I interferonopathy state and a previously characterised type I

interferonopathy associated with mutations in R99 of ARF1. Thus, we aimed to elucidate the effect of ARF1 R19C on IFN signalling.

In line with elevated ISG expression in patient blood, and similar to supernatants of fibroblasts heterozygous for ARF1 R99C, supernatants of R19C patient-derived (AGS3238) fibroblasts induced type I IFN signalling in reporter cells (293-Dual hSTING-R232) (Fig. 1B). To understand if the R19C mutation is responsible for enhanced type I IFN signalling, we constructed expression vectors. Expression of ARF1 R19C promoted IFN signalling in a dose-dependent manner in HEK293T cells only upon co-expressing STING. As expected, in the absence of STING co-expression, neither R19C nor R99C showed IFN induction (Figs. 1C and EV1A). Dose-dependency analysis revealed that ARF1 R19C expression led to a delayed onset of IFN signalling compared to R99C, but reached higher signalling levels (Figs. 1D and EV1B). Expression of ARF1 WT did not induce type I IFN signalling at any expression level.

In summary, these findings suggest that ARF1 R19C promotes aberrant type I IFN signalling dependent on STING.

### ARF1 R99C, but not R19C, promotes mtDNA release

We recently showed that mutation ARF1 R99C compromises mitochondrial integrity thus promoting mtDNA release into the cytoplasm and providing a trigger for cGAS (Hirschenberger et al, 2023). To investigate the impact of the R19C mutation on mitochondrial integrity, primary fibroblasts from patient AGS3238 (R19C), AGS460 (R99C) or a healthy control (WT) were subjected to electron microscopy. The mitochondria in ARF1 WT and R19C fibroblast samples exhibited an intact phenotype with no discernible differences in the distance between the inner and outer mitochondrial membrane and the appearance of cristae, while the integrity of the mitochondria in ARF1 R99C fibroblasts was visibly disrupted (Fig. 2A). mtDNA levels in the cytoplasm upon expression of ARF1 R19C in HEK293T were comparable or even lower than the WT or vector control (Figs. 2B,C and EV1C). In contrast, expression of ARF1 R99C, or treatment with ABT-737 and Q-VD-OPH (ABT/QVD), which serve as positive controls to induce mtDNA release, led to a significant increase in cytosolic mtDNA levels. Mitochondrial membrane potential was not altered in ARF1 R19C expressing HEK293T cells compared to WT ARF1 expressing cells (Fig. EV1D). Of note, in 293-Dual hSTING-R232 cells, which were engineered to constitutively express human STING, ARF1 R19C failed to stimulate ISRE-SEAP reporter activity in contrast to ARF1 R99C (Figs. 2D and EV1E). This indicates that expression of ARF1 R19C is not sufficient to induce type I IFN signalling unlike ARF1 R99C. To analyse the impact of the ARF1 R19C mutation in primary fibroblasts we compared ISG induction in cells from patient AGS3238 (R19C) and three healthy donors (WT 1, 2, 3) upon treatment with cGAMP. The mRNA levels of the ISGs Mx1 and OAS1, which exhibited elevated expression in the blood of AGS3238 (Fig. 1A), were quantified using qPCR. The mRNA levels of Mx1 increased by ~200-fold in ARF1 WT fibroblasts and ~600-fold in fibroblasts with the R19C mutation (Fig. 2E). Similarly, the levels of OAS1 showed a ~1600-fold increase in healthy donor fibroblasts and a ~5600-fold increase in fibroblasts from patient AGS3238 (R19C) (Fig. 2F). Thus, the expression of both ISGs was consistently approximately 3-fold

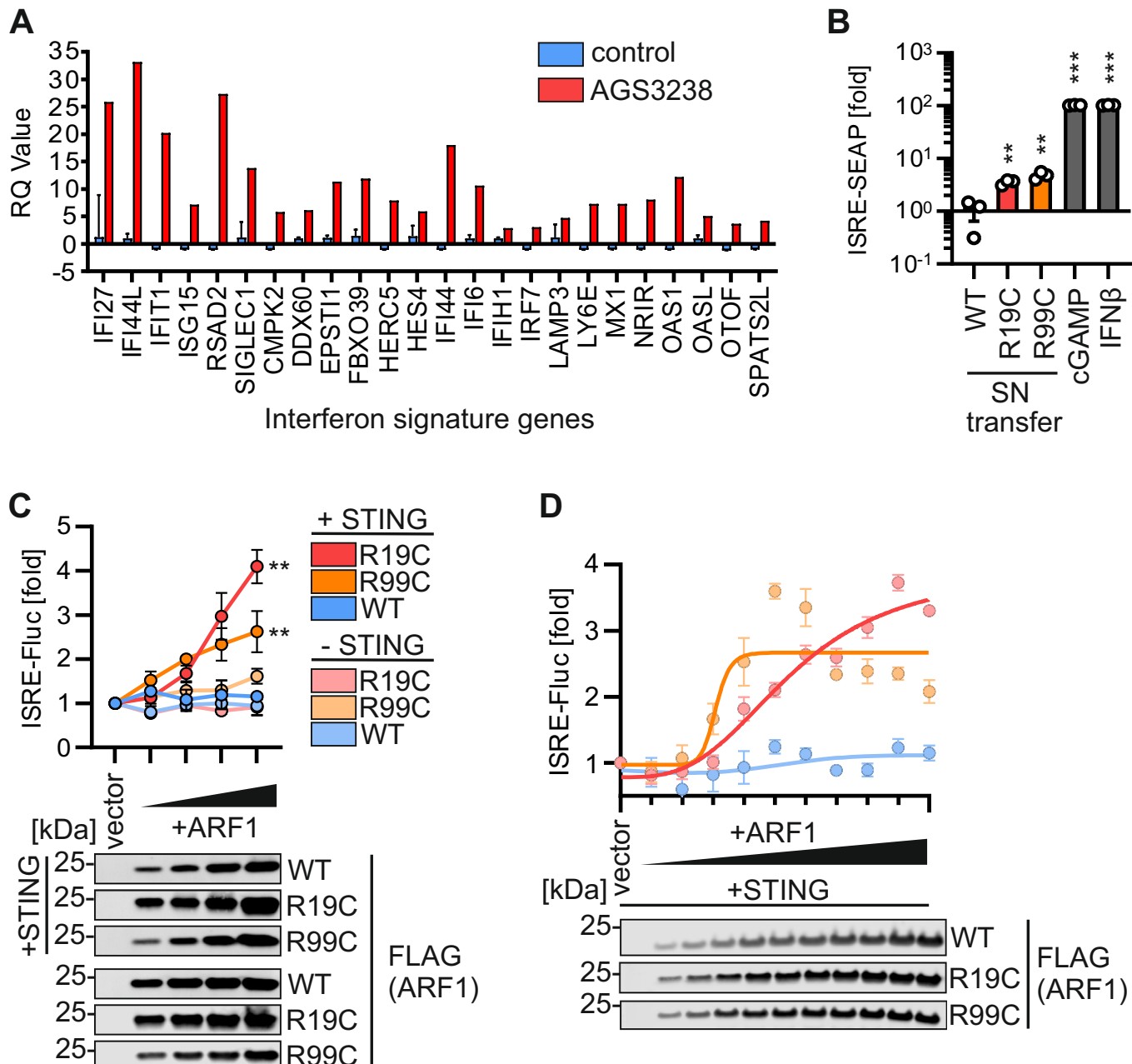

**Figure 1. ARF1 R19C induces IFN signalling dependent on STING.**

(A) Interferon signature gene expression of 24 ISGs in PBMCs of patient AGS3238 (red) compared to the average of 29 healthy donors (blue). $n = 1$ (patient), $n = 29$ (healthy donors) ± SD. (B) Supernatant (SN) transfer from primary fibroblasts of a healthy donor (WT), patient AGS3238 (R19C) or patient AGS460 (R99C) to HEK293-STING cells (293-Dual-hSTING-R232). IFN-β (100 U/mL, 48 h) and cGAMP (10 μg/ml, 48 h) were used as positive controls. ISRE-SEAP activity was quantified 48 h (h) post transfer and normalised to metabolic activity. Bars present the mean of $n = 3$ ± SEM (biological replicates). Statistical analysis was performed using two-tailed Student's t test with Welch's correction. **$p < 0.01$ ($p = 0.0059$ WT vs R19C, $p = 0.0029$ WT vs R99C); ***$p < 0.001$ ($p < 0.0001$ WT vs cGAMP, WT vs IFN-β). (C) ISRE-Firefly luciferase (Fluc) activity of HEK293T cells transiently expressing increasing amounts of FLAG-tagged ARF1 WT, R19C or R99C, together with STING ($+$ STING) or vector control (-STING), quantified 39 h post transfection and normalised to metabolic activity. Dots present the mean of $n = 3$ ± SEM (biological replicates). Statistical analysis of the area under the curve was performed using two-tailed Student's t test with Welch's correction. **$p < 0.01$ ($p = 0.0041$ WT vs R19C (both $+$STING), $p = 0.0038$ WT vs R99C (both $+$STING)). Representative immunoblots were stained with anti-FLAG. (D) ISRE-Fluc activity of HEK293T cells transiently expressing increasing amounts (10–100 ng, in steps of 10 ng) of FLAG-tagged ARF1 WT, R19C or R99C, together with STING quantified 37 h post transfection and normalised to metabolic activity. Nonlinear regression curves are highlighted. Dots present the mean of $n = 3$ ± SEM (biological replicates). Representative immunoblots were stained with anti-FLAG. Source data are available online for this figure.

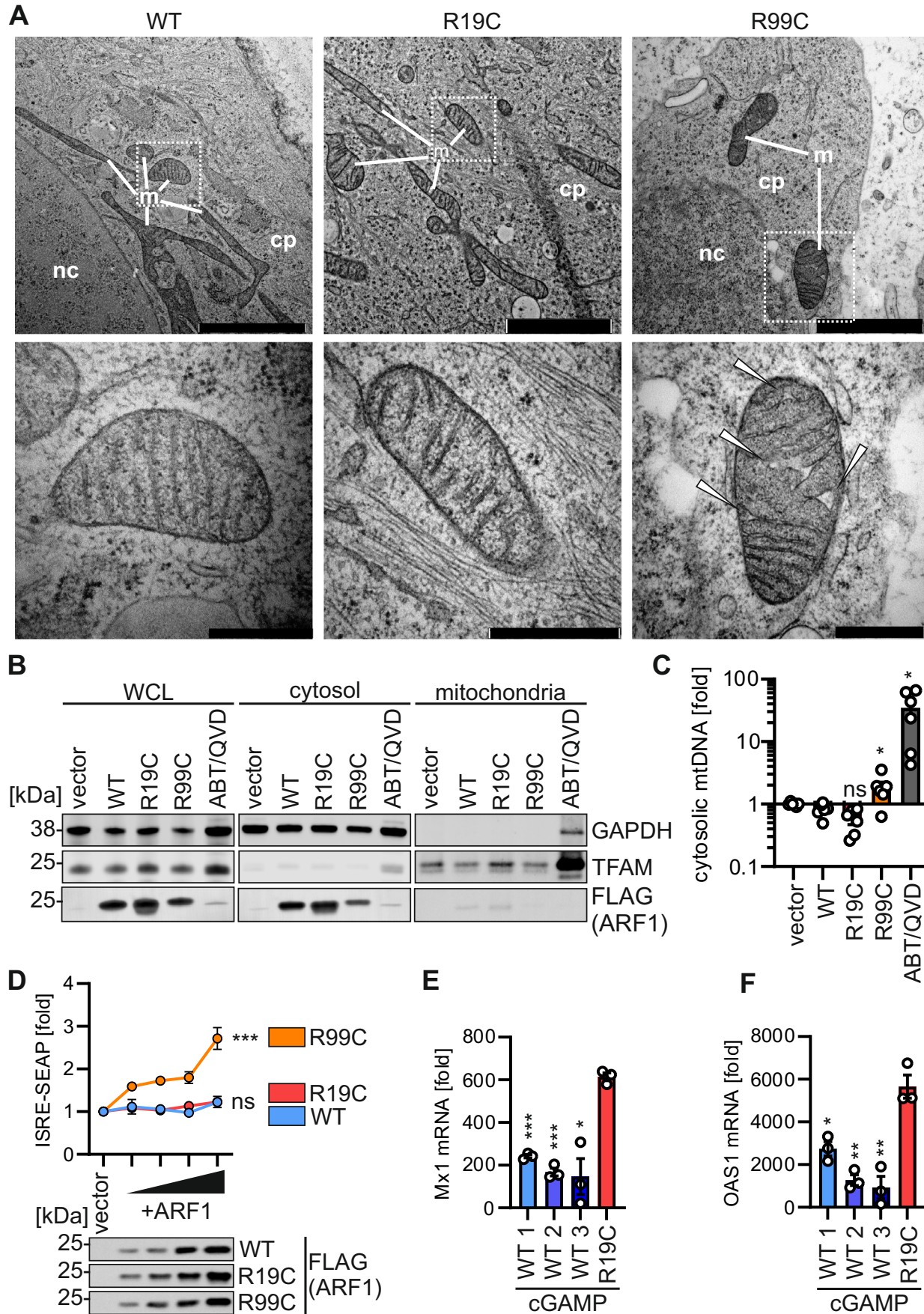

◄

**Figure 2. ARF1 R19C does not promote the release of mtDNA.**

(A) Representative transmission electron microscopy images of primary fibroblasts of a healthy donor (WT), patient AGS3238 (R19C) or AGS460 (R99C). Scale bar: 2 μm (overview images), 200 nm (higher magnification image). Inflated cristae, white arrows. cp, cytoplasm. m, mitochondria. nc, nucleus. (B) Representative immunoblots showing fractionation of HEK293T cells transiently expressing FLAG-tagged ARF1 WT, R19C, R99C, or vector control or treated with ABT-737/Q-VD-OPH (10 μM each, 24 h). Blots were stained with anti-FLAG, anti-TFAM and anti-GAPDH. (C) qPCR analysis of mitochondrial (mt) DNA (MT-D-Loop) in the cytosolic fraction of (B), normalised to total cellular mtDNA (MT-D-Loop/KCNJ10). Bars represent the mean of $n = 6 \pm$ SEM (biological replicates). Statistical analysis was performed using two-tailed Student's t test with Welch's correction. $*p < 0.05$ ($p = 0.0471$ WT vs R99C, $p = 0.0289$ WT vs ABT/QVD); ns, not significant ($p = 0.0763$ WT vs R19C). (D) ISRE-SEAP activity of HEK293-STING cells transiently expressing increasing amounts of FLAG-tagged ARF1 WT, R19C or R99C, quantified 37 h post transfection and normalised to metabolic activity. Dots present the mean of $n = 3 \pm$ SEM (biological replicates). Statistical analysis of the area under the curve was performed using two-tailed Student's t test with Welch's correction. $***p < 0.001$ ($p = 0.0004$ WT vs R99C); ns, not significant ($p = 0.6121$ WT vs R19C). Representative immunoblots were stained with anti-FLAG. (E, F) fold induction of Mx1 (E) and OAS1 (F) mRNA levels in primary fibroblasts from healthy donors (WT 1, 2, 3) or patient AGS3238 (R19C) as assessed by qPCR 16 h post stimulation with cGAMP (20 μg/ml). Lines represent the mean of $n = 3 \pm$ SEM (biological replicates). Statistical analysis was performed using two-tailed Student's t test with Welch's correction. $*p < 0.05$ ($p = 0.0267$ WT3 vs R19C (E), $p = 0.016$ WT1 vs R19C (F)); $**p < 0.01$ ($p = 0.0067$ WT2 vs R19C (F), $p = 0.0033$ WT3 vs R19C (F)); $***p < 0.001$ ($p = 0.0005$ WT1 vs R19C (E), $p < 0.0001$ WT2 vs R19C (E)). Source data are available online for this figure.

higher in patient-derived fibroblasts as compared to ARF1 WT fibroblasts upon stimulation with cGAMP.

In summary, our data suggest that the R19C mutation does not cause mitochondrial instability. While ARF1 R99C compromises mitochondrial integrity by increasing levels of mitofusin 1 (MFN1) and promoting mitochondrial hyperfusion (Hirschenberger et al, 2023), which results in the release of mtDNA into the cytoplasm, the R19C mutation does not interfere with mitochondrial integrity (Figs. 2A–C and EV1C), and therefore does not promote activation of cGAS by self-derived DNA. However, after initial STING activation, ARF1 R19C, similar to ARF1 R99C, promotes type I IFN signalling. A dispensable role of cGAS activation was previously proposed for COPA mutations that led to chronically enhanced type I interferon signalling (Mukai et al, 2021; Kemmoku et al, 2024).

## Golgi morphology and retrograde transport are affected by ARF1 R19C

A second function of ARF1 in the cGAS-STING pathway is COPI-dependent retrograde transport of STING from the Golgi to the ER, to promote STING deactivation (Hirschenberger et al, 2023). Our previous data showed that ARF1 R99C has impaired shuttling capability due to decreased binding to COPI vesicles and reduced GTPase activity. This results in increased ERGIC/Golgi localisation and disturbed Golgi morphology (Hirschenberger et al, 2023). In line with this, ARF1 R19C also displays increased ERGIC co-localisation (Fig. EV2A,B). Electron microscopy unveiled that healthy donor fibroblasts display a smooth Golgi membrane profile with well-defined stacks and a parallel orientation of cisternae (Fig. 3A, left panel and EV2C). Conversely, the morphology of the Golgi stacks is visibly altered in all examined primary patient fibroblasts (ARF1 R19C), characterized by disordered cisternae, and signs of Golgi fragmentation (Figs. 3A and EV2C). Quantification of the electron microscopy images confirmed a higher number of fragmented Golgi in R19C patient fibroblasts (68%), compared to healthy control fibroblasts (25%, Fig. 3B). To understand whether ARF1 R19C is unable to interact with COPI, we analysed binding of ARF1 R19C to γ-COP, a principal component of COPI vesicles, that was previously associated with ARF1 (Yu et al, 2012). Immunoprecipitation experiments showed a reduction in the binding of endogenous γ-COP to ARF1 R19C compared to WT ARF1 in HEK293T cells overexpressing FLAG-tagged ARF1

variants (Fig. 3C). Endogenous ARF1 readily co-precipitated γ-COP in healthy donor fibroblasts (Fig. EV2D). A γ-COP-ARF1 interaction in primary ARF1 R19C patient fibroblasts could not be detected, however, this was potentially occluded by very low basal γ-COP levels in the patient fibroblasts. GTPase activity of ARF family proteins is required for trafficking (Hirschenberger et al, 2023; Sztul et al, 2019). Thus, we wondered whether a mutation at position R19 would impact GTP turnover. In vitro GTPase assay revealed that ARF1 R19C exhibited significantly reduced GTPase activity, comparable to that of ARF1 Q71L, a variant known to lack GTPase activity (Fig. 3D). Structural analysis suggests that positioned on ß-sheet 1 (Fig. EV2E), R19 is unlikely to directly engage in GTP-binding (Ménétrey et al, 2007). Nevertheless, it has been suggested that R19 plays a central role in the GDP-GTP switch of ARF1 (Agathe et al, 2023), and substitutions at this residue may interfere with GTPase function.

To experimentally evaluate whether impaired γ-COP binding and GTPase activity of R19C have an impact on ARF1-mediated Golgi-to-ER trafficking, retrograde transport was analysed in HeLa cells expressing a thermolabile eGFP-tagged VSV-G protein containing the KDEL retrograde trafficking signal (VSVGtsO45-KDELR) (Cole et al, 1998). Temperature shifts from 37 °C to 32 °C induced Golgi translocation of the fusion protein, which is usually localized at the ER, while subsequent shifts from 32 °C to 40 °C prompt ER relocation (Fig. EV3A). Colocalization between VSV-G and the cis-Golgi marker GM130 serves as a readout for transport. Expression of ARF1 WT, R19C, or R99C had no discernible effect on anterograde transport, as reflected by an increased colocalization of VSV-G with the cis-Golgi upon temperature reduction (Fig. 3E). However, upon elevating the temperature to 40 °C, while unaffected by ARF1 WT expression, retrograde transport was almost entirely abolished in the presence of ARF1 R19C or R99C (Fig. 3E).

Taken together, ARF1 R19C exhibits decreased GTPase activity and interaction with γ-COP, resulting in disturbed Golgi integrity and impaired retrograde transport. This observation suggests that the impact of ARF1 on mitochondrial stability is not dependent on the GTPase function, but rather driven by other factors, e.g., a disturbance of protein interactions. For instance, it has been proposed that ARF1 regulates mitochondrial dynamics through the recruitment of VCP to mitochondria (Hirschenberger et al, 2023; Ackema et al, 2014), although a direct interaction has not yet been demonstrated. Similar to ARF1 R99C, ARF1 R19C displays a

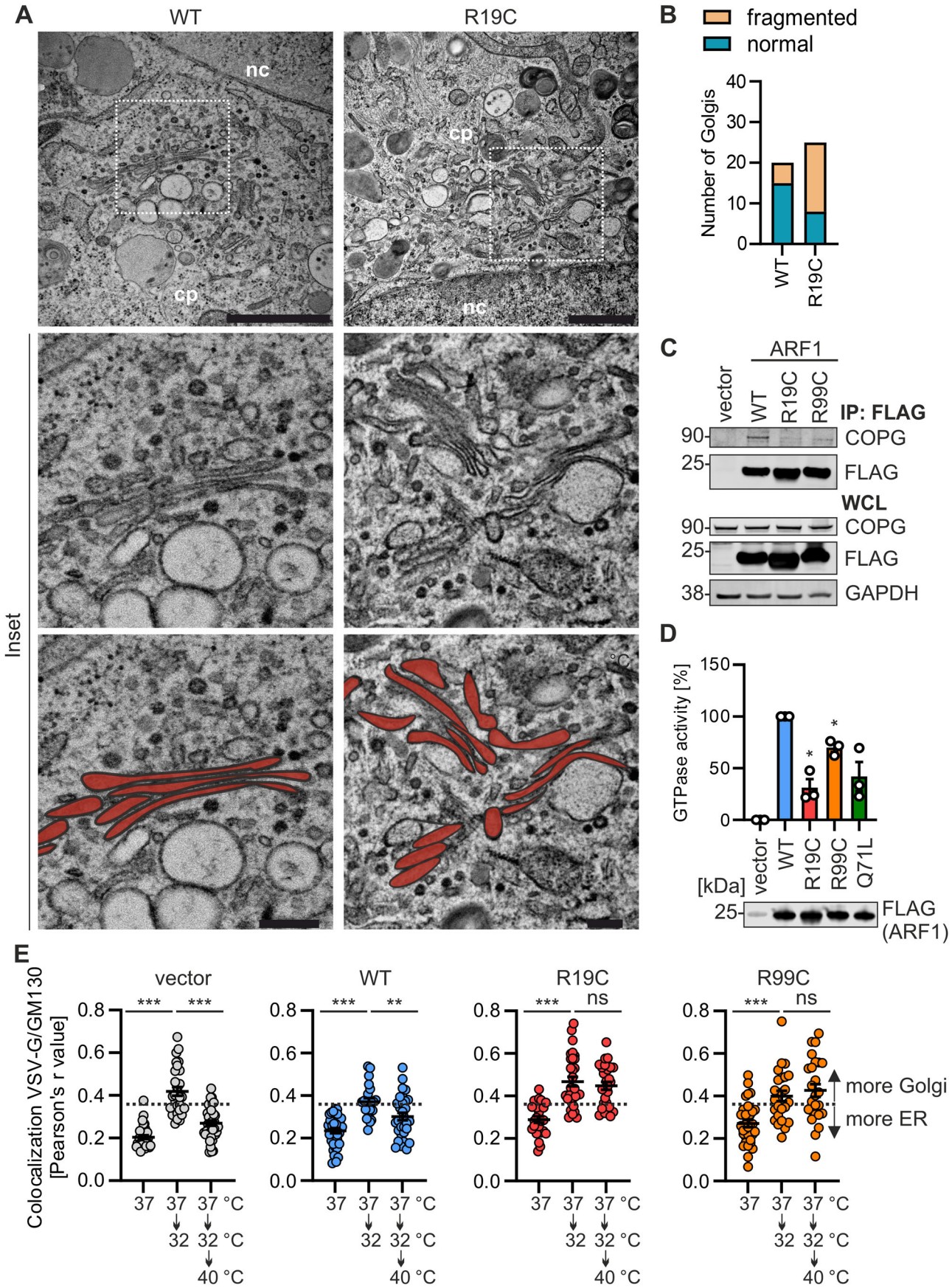

◄ **Figure 3. ARF1 R19C shows decreased interaction with COPG and impairs retrograde transport.**

(A) Representative transmission electron microscopy images of the Golgi area of primary fibroblasts from a healthy donor (WT) or patient AGS3238 (R19C). Scale bar: 1 μm (overview images), 200 nm (higher magnification image). Stacked Golgi cisternae are highlighted in red. cp, cytoplasm. nc, nucleus. (B) Quantification of Golgi disorganisation (normal, fragmented) from the images shown in (A). $n = 20$–25 (individual cells). (C) Immunoprecipitation (IP) of WCLs of HEK293T cells expressing FLAG-tagged ARF1 WT, R19C, R99C or vector control using anti-FLAG coupled dynabeads. Immunoblots of the WCL and the IP were stained with anti-COPG, anti-FLAG and anti-GAPDH. (D) GTPase activity of ARF1 WT, R19C, R99C and Q71L purified from HEK293T cells, normalised to WT ARF1. Bars present the mean of $n = 3 \pm$ SEM (biological replicates). Statistical analysis was performed using two-tailed Student's t test with Welch's correction. *$p < 0.05$ ($p = 0.0144$ WT vs R19C, $p = 0.0185$ WT vs R99C). Representative immunoblots were stained with anti-FLAG. (E) Colocalization (Pearson's correlation coefficient, r) between VSVG-ts045-KDELR (VSV-G) and GM130 in HeLa cells transiently expressing ARF1 WT, R19C, R99C or vector control together with VSV-G. Temperature shifts (37 °C/32 °C/40 °C) are indicated. Lines represent the mean of $n = 22$–40 $\pm$ SEM (individual cells). Statistical analysis was performed using two-tailed Student's t test with Welch's correction. **$p < 0.01$ ($p = 0.0056$ WT (37 °C/32 °C/40 °C)); ***$p < 0.001$ ($p < 0.0001$ vector, WT, R19C, R99C (37 °C/32 °C), vector (37 °C/32 °C/40 °C)); ns, not significant ($p = 0.4966$ R19C, $p = 0.5004$ R99C (37 °C/32 °C/40 °C)). Source data are available online for this figure.

diminished interaction with γ-COP, disrupting the transport of STING from the Golgi back to the ER (Fig. 3C,E) (Hirschenberger et al, 2023). Consequently, overexpression of ARF1 R19C does not induce IFN signalling in 293-Dual hSTING-R232 cells, which express STING (Fig. 2D), as opposed to ARF1 R99C and ARF1 Q71L, that both promote mtDNA release (Hirschenberger et al, 2023). Of note, steady-state γ-COP levels in ARF1 R19C patient cells were diminished compared to healthy donor fibroblasts. This may be due to increased turnover of accumulated, aberrant COPI vesicles/Golgi fragments; however, the precise reason needs further investigation.

## ARF1 R19C accumulates STING at the Golgi and activates TBK1

To promote termination of cGAS-STING signalling, STING is, besides degradation via lysosomes, recycled in an ARF1-dependent manner from the Golgi to the ER. Thus, we examined the colocalization of transiently expressed STING-eGFP and GM130 in HeLa cells expressing ARF1 WT, R19C, or R99C. While WT ARF1 did not alter STING localization, both ARF1 R19C and ARF1 R99C led to a significant increase in the colocalization of STING-eGFP with GM130 (Fig. 4A,B). To recapitulate a more physiological setting, we used primary fibroblasts isolated from patients AGS3238 (R19C) and AGS460 (R99C) and compared them to healthy control fibroblasts (WT). Endogenous STING displayed an enhanced localization at the cis-Golgi in fibroblasts from patients with the ARF1 R19C or R99C mutation, comparable to cGAMP-treated healthy cells (Fig. 4C,D). Additionally, the colocalization of STING with the ERGIC marker ERGIC-53 was elevated in fibroblasts derived from these patients (Fig. EV3B,C). To assess downstream signalling initiated by Golgi-localized STING, we quantified phospho-TBK1 (pTBK1, p-S172) levels as a percentage of the cell area. In primary fibroblasts from patient AGS3238 (R19C) the levels of pTBK1 were significantly increased compared to healthy (WT) control fibroblasts (Fig. 4E,F). In line with this, analysis of endogenous S-172 phosphorylation of TBK1 by western blotting, revealed higher levels of pTBK1 6 h post stimulation with cGAMP in primary patient fibroblasts compared to healthy control fibroblasts (Fig. EV3D).

Collectively, these findings provide evidence that STING accumulates at the ERGIC/Golgi in the presence of ARF1 R19C, thereby enhancing TBK1 activation and downstream signalling. While the previously described ARF1 R99C mutation disturbs mitochondrial integrity and retrograde transport (Hirschenberger et al, 2023), ARF1 R19C solely impairs retrograde trafficking.

Consequently, this mutation provides a valuable opportunity for investigating the dual role of ARF1 in regulating innate immunity.

## Innate immune activation is prolonged by ARF1 R19C

Increased colocalization of STING with the cis-Golgi marker GM130 (Fig. 5A,B), along with elevated levels of pTBK1 as analysed by immunofluorescence imaging and western blot, in patient AGS3238 (R19C) fibroblasts compared to healthy control fibroblasts 4 to 24 h post-stimulation, suggest prolonged IFN signalling (Figs. 5C and EV3D, EV4A). To investigate this in more detail, the mRNA levels of Mx1 and OAS1 were followed up to five days post cGAMP stimulation. In healthy donor fibroblasts, the mRNA levels of Mx1 and OAS1 (Fig. 5D,E) peaked one day post-treatment and then declined. In contrast, fibroblasts from the ARF1 R19C patient not only showed a higher initial induction, they also declined more slowly compared to healthy fibroblasts (Fig. 5D,E). Of note, polyI:C stimulation led to a similar induction of ISG levels in ARF1 WT and R19C fibroblast at an early (Day 1) and at a late timepoint after stimulation (Day 5), suggesting the differences in decline are specific to cGAS-STING signalling (Fig. EV4B). Recently, it was proposed that STING turnover by autophagy may impact sustained STING signalling. To analyse whether ARF1 R19C alters autophagic flux and subsequent STING degradation, we monitored endogenous LC3B-I to II conversion and endogenous STING levels by western blotting in primary human fibroblasts. Treatment of the primary fibroblasts with Bafilomycin A1, which inhibits autophagic flux, and Torin-1, which induced autophagy showed that both WT and ARF1 R19C fibroblasts display similar LC3B processing efficiencies (Fig. EV4C). This suggests that autophagic flux is not altered in the presence of ARF1 R19C. Consequently, STING degradation after cGAMP stimulation, occurred with similar kinetics in WT ARF1 and R19C patient cells (Fig. EV4D).

Overall, these results indicate that IFN signalling is not only elevated initially, but also prolonged in the presence of ARF1 R19C. This suggests that DNA sensing termination pathways are impaired. Of further note, patients harbouring either the ARF1 R19C or the R99C variant exhibited enhanced ISG expression in blood on the occasions tested, but only the ARF1 R99C variant provides a cGAS self-ligand. In line with this, our data shows that expression of ARF1 R19C, even in the presence of STING and cGAS, does not active type I IFN signalling (Fig. 2D). Accumulation of STING at the ERGIC/Golgi in ARF1 R19C patient-derived fibroblasts occurs even without prior stimulation, along with elevated pTBK1 levels (Figs. 4A–F and EV3B,C). This is similar to patients with defects in COPA (Deng et al, 2020; Lepelley et al,

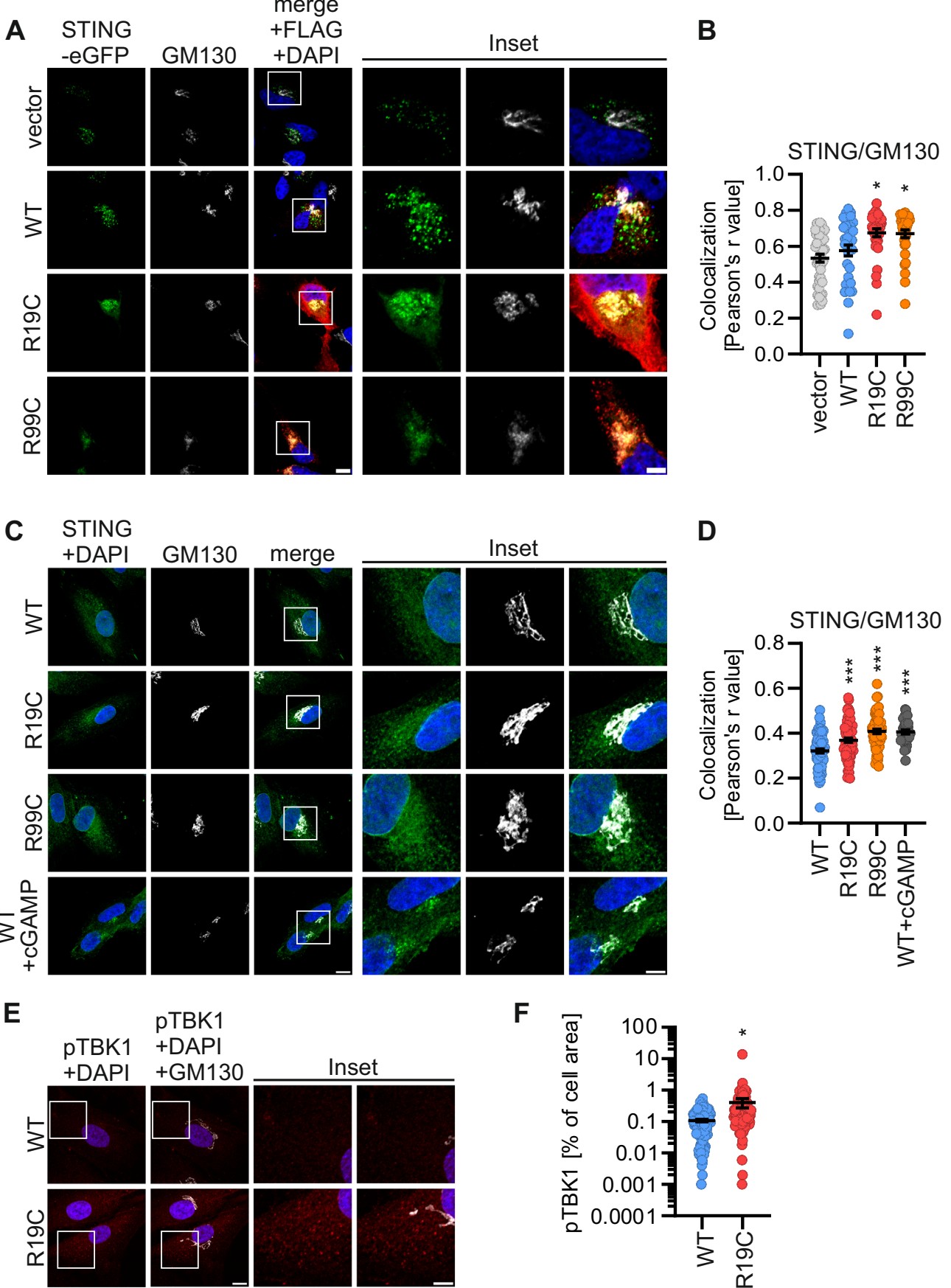

**Figure 4. STING accumulates at the Golgi in the presence of ARF1 R19C.**

(A) Representative immunofluorescence images of Hela cells expressing ARF1 WT, R19C, R99C or vector control together with STING-eGFP (green). 24 h post transfection the cells were stained with anti-FLAG (red) and anti-GM130 (white). Nuclei: DAPI (blue). Insets are shown in higher magnification. Scale bar: 10 μm (full size images), 5 μm (insets). (B) Colocalization (Pearson's correlation coefficient, r) between STING and GM130 from the images shown in (A). Lines represent the mean of $n = 32$–$42 \pm$ SEM (individual cells). Statistical analysis was performed using two-tailed Student's t test with Welch's correction. *$p < 0.05$ ($p = 0.011$ WT vs R19C, $p = 0.0153$ WT vs R99C). (C) Representative immunofluorescence images of primary fibroblasts of a healthy donor (WT), patient AGS3238 (R19C) or patient AGS460 (R99C). Healthy donor fibroblasts treated with cGAMP (20 μg/ml, 2 h) were used as positive control. Cells were stained with anti-STING (green) and anti-GM130 (white). Nuclei: DAPI (blue). Insets are shown in higher magnification. Scale bar: 10 μm (full size images), 5 μm (insets). (D) Colocalization (Pearson's correlation coefficient, r) between STING and GM130 from the images shown in (C). Lines represent the mean of $n = 38$–$71 \pm$ SEM (individual cells). Statistical analysis was performed using two-tailed Student's t test with Welch's correction. ***$p < 0.001$ ($p < 0.0001$ WT vs R99C, WT vs cGAMP, $p = 0.0003$ WT vs R19C). (E) Representative immunofluorescence images of primary fibroblasts of a healthy donor (WT) or patient AGS3238 (R19C) 24 h post seeding. Cells were stained with anti-pTBK1 (red) and anti-GM130 (white). Nuclei: DAPI (blue). Insets are shown in higher magnification. Scale bar: 10 μm (full size images), 5 μm (insets). (F) Quantification of pTBK1 puncta (% of area occupied by TBK1 signal per cell) from the images shown in (E). Lines represent the mean of $n = 104$–$115 \pm$ SEM (individual cells). Statistical analysis was performed using two-tailed Student's t test with Welch's correction. *$p < 0.05$ ($p = 0.0282$ WT vs R19C). Source data are available online for this figure.

2020; Mukai et al, 2021; Steiner et al, 2022; Kemmoku et al, 2024). It is tempting to speculate that external forces like DNA virus infection may serve as the initial trigger for ARF1 R19C associated IFN overexpression and lead to a basal chronic level of type I IFN-associated inflammation. Notably, the patient-derived PBMCs show a robust IFN signature, while fibroblasts isolated from the same patient do not display an apparent upregulation of ISGs. PBMCs contain many immune-related cells that may react more sensitively towards the presence of type I IFN. In addition, PBMCs may have experienced more previous activation events than fibroblasts that eventually led to sustained type I IFN release. However, tonic activity of the cGAS-STING pathway may also result in chronic inflammation if not appropriately terminated (Chu et al, 2021; Tu et al, 2022; Lepelley et al, 2020; Steiner et al, 2022; Klute et al, 2022). Thus, a defect in negative regulation, in the absence of an internal trigger i.e. self-DNA, may be sufficient to induce auto-inflammation. Of note, R19 of ARF1 has been reported to interact with the γ-subunit of the AP-1 complex (Ren et al, 2013; Agathe et al, 2023), which is involved in cargo sorting from the trans-Golgi to endosomes and plays a role in STING signalling termination (Jeltema et al, 2023; Liu et al, 2022). The exact role of AP-1 in this context is unclear, as different studies indicate both diminished (Fang et al, 2023) and enhanced (Liu et al, 2022) immune activation following AP-1 knockdown. Notably, STING expression levels were not altered in the presence of ARF1 R19C (Figs. 1D and EV1A,B,E) and STING degradation or autophagy is not impaired in patient fibroblasts (Fig. EV4C,D), indicating that AP-1 mediated STING degradation is not affected by ARF1 R19C.

## H-151 and amlexanox treatment reduce chronic IFN production in patient cells in vitro

To date, treatment of type I interferonopathies is insufficient. To confirm that ARF1 R19C dysregulates type I IFN signalling specific to the cGAS-STING signalling pathway, we depleted STING from primary patient fibroblasts and healthy controls using Cas9/gRNA RNP electroporation. Western blot analysis showed that endogenous levels of STING were strongly reduced in healthy control fibroblasts and patient AGS3238 (R19C) fibroblasts by Cas9/gRNA treatment (Fig. 5F). Quantification of the mRNA levels of the ISGs Mx1, OAS1 and RSAD2 using qPCR indicated that STING was required for cGAMP signalling, both in ARF1 WT and R19C fibroblasts (Figs. 5F and EV5A). As expected, induction of ISGs by

the RIG-I agonist polyI:C and the TLR4 agonist LPS was not dependent on STING (Fig. EV5B). To investigate a putative therapeutic potential of compounds targeting type I IFN signalling in vitro, we employed the STING palmitoylation inhibitor H-151 (Haag et al, 2018) and the TBK1 inhibitor amlexanox (Reilly et al, 2013). Both inhibitors demonstrated a dose-dependent decrease in IFN signalling in cGAMP-stimulated reporter cells (Fig. EV5C). Subsequently, fibroblasts from a healthy control donor and patients carrying the R19C or R99C mutation underwent treatment with these inhibitors every two days over an 8-day period (four treatments in total). Notably, treatment with H-151 or amlexanox significantly reduced IFN signalling in fibroblasts from patient AGS3238 (R19C), by about 8.0-fold and 2.4-fold, respectively (Fig. 5G), with only minimal cytotoxic effects (Fig. EV5E). In the case of R99C fibroblasts, both inhibitors led to a roughly 2.8-fold decrease in ISRE reporter activity, albeit accompanied by moderate cytotoxic effects (Figs. 5H and EV5E). As expected, in healthy donor cells none of the treatments led to a decrease of IFN signalling, as assessed by supernatant transfer to ISRE reporter cells (Fig. EV5D,E).

In summary, our findings suggest that treatment with the STING inhibitor H-151 or the TBK1 inhibitor amlexanox reduce IFN signalling in ARF1-mutant primary patient cells ex vivo.

Although IFNs are considered to be central to the pathogenesis of monogenic states currently classified as type I interferonopathies, it remains uncertain whether the presence of chronic IFN is the sole cause for the clinical features observed (Crow and Stetson, 2022). In the context of ARF1, both ARF1 R19C and ARF1 R99C mutations have been associated with elevated ISG expression in blood, and with in vitro and ex vivo evidence of dysregulated IFN signalling. Of note, while the current study involves a single patient with ARF1 R19C, the skin diseases seen in certain patients with the R99C mutation were not present (Hirschenberger et al, 2023). Further, the R19C patient described here presented with periventricular nodular heterotopia (PVNH), a feature previously described in other patients with ARF1-related disease. Notably, abnormalities of neuronal migration are not a characteristic of other type I interferonopathies, leading us to suspect that this feature is not IFN-related. Indeed, ARF1-mediated retrograde trafficking is crucial not only for maintaining STING homeostasis but also for the return of KDEL-containing proteins, such as ER chaperones (Yamamoto et al, 2001). Consequently, COPI-deficient cells exhibited increased localization of KDEL at the Golgi (Steiner

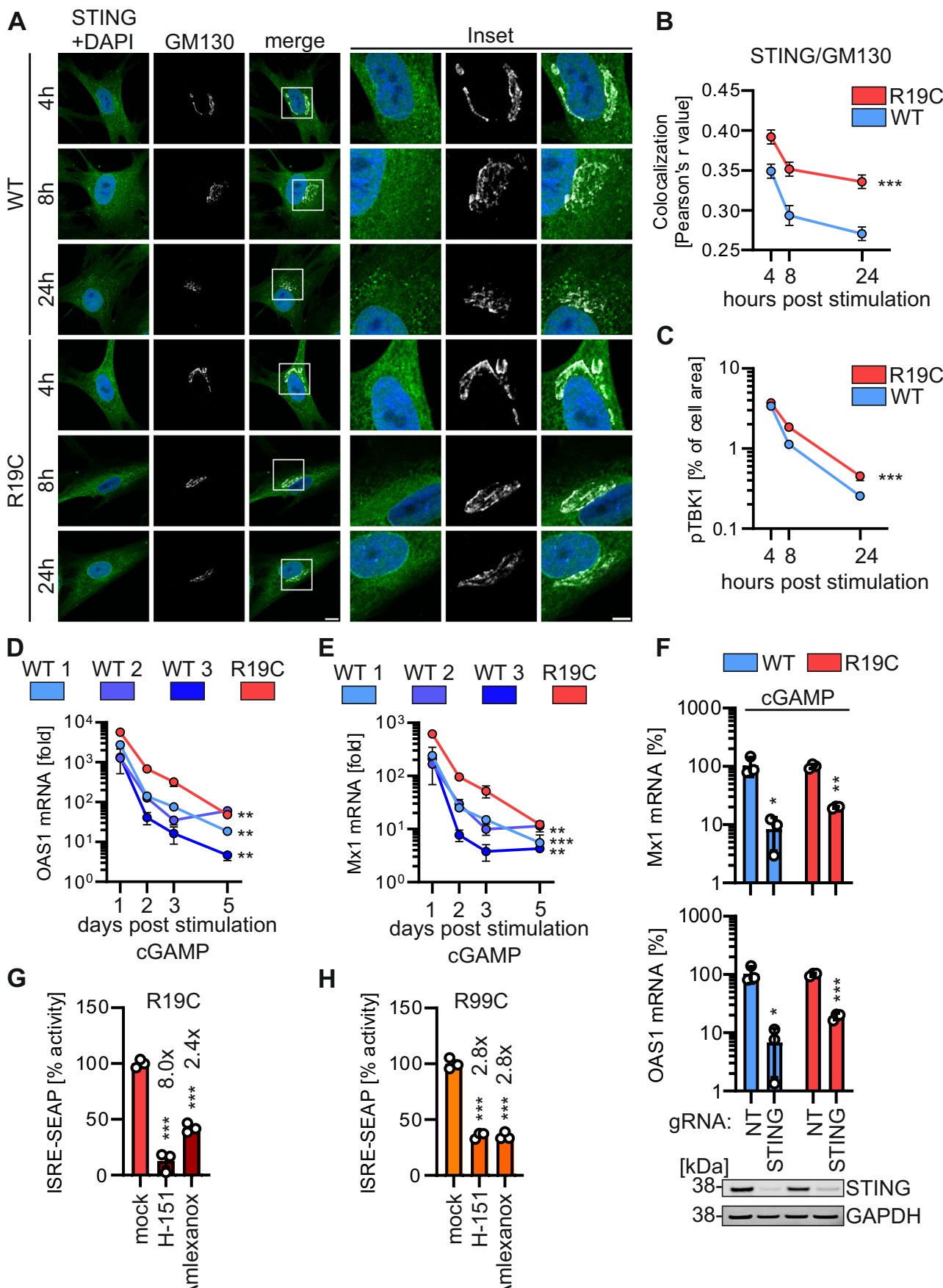

◄ **Figure 5. Innate immune activation is prolonged in ARF1 R19C expressing cells and limited by treatment approaches.**

(A) Representative immunofluorescence images of primary fibroblasts of a healthy donor (WT) or patient AGS3238 (R19C). Cells were treated with cGAMP (50 µg/ml) and stained at the indicated timepoints post stimulation with anti-STING (green) and anti-GM130 (white). Nuclei: DAPI (blue). Insets are shown in higher magnification. Scale bar: 10 µm (full size images), 5 µm (insets). (B) Colocalization (Pearson's correlation coefficient, r) between STING and GM130 from the images shown in (A). Lines represent the mean of $n = 67–81 \pm$ SEM (individual cells). Statistical analysis of the area under the curve was performed using two-tailed Student's t test with Welch's correction. ***$p < 0.001$ ($p < 0.0001$ WT vs R19C). (C) Quantification of pTBK1 puncta (% of area occupied by TBK1 signal per cell) in primary fibroblasts of a healthy donor (WT) or patient AGS3238 (R19C) via immunofluorescence microscopy at indicated time points following stimulation with cGAMP (50 µg/ml). Lines represent the mean of $n = 100–139 \pm$ SEM (individual cells). Statistical analysis of the area under the curve was performed using two-tailed Student's t test with Welch's correction. ***$p < 0.001$ ($p < 0.0001$ WT vs R19C). (D, E) fold induction of OAS1 (D) or Mx1 (E) mRNA levels in primary fibroblasts from healthy donors (WT 1, 2, 3) or patient AGS3238 (R19C) as assessed by qPCR at indicated time points following stimulation with cGAMP (20 µg/ml). Lines represent the mean of $n = 2–3 \pm$ SEM (biological replicates). Statistical analysis of the area under the curve was performed using two-tailed Student's t test with Welch's correction. **$p < 0.01$ ($p = 0.0046$ WT1 vs R19C, $p = 0.0032$ WT2 vs R19C, $p = 0.0059$ WT3 vs R19C (D), $p = 0.0018$ WT1 vs R19C $p = 0.0048$ WT3 vs R19C (E)); ***$p < 0.001$ ($p = 0.0002$ WT2 vs R19C (E)). (F) fold induction of Mx1 (top) and Oas1 (bottom) mRNA levels in primary fibroblasts from a healthy donor (WT) or patient AGS3238 (R19C) as assessed by qPCR 16 h post stimulation with cGAMP (20 µg/ml). Treatment was performed 96 h post electroporation with Cas9/gRNA RNPs targeting either STING or non-targeting (NT). Lines represent the mean of $n = 3 \pm$ SEM (biological replicates). Statistical analysis was performed using two-tailed Student's t test with Welch's correction. *$p < 0.05$ ($p = 0.0445$ NT vs STING (WT, Mx1), $p = 0.0334$ NT vs STING (WT, Oas1)); **$p < 0.01$ ($p = 0.0035$ NT vs STING (R19C, Mx1); ***$p < 0.001$ ($p = 0.0004$ NT vs STING (R19C, Oas1). Representative immunoblots were stained with anti-STING and anti-GAPDH. (G, H) Supernatant transfer from primary fibroblasts of patient AGS3238 (R19C, G) or patient AGS460 (R99C, H) consecutively treated four times every 48 h with H-151 (2 µM) or amlexanox (33 µg/ml) to HEK293-STING cells (293-Dual-hSTING-R232). ISRE-SEAP activity was quantified 48 h post transfer and normalised to metabolic activity. Lines represent the mean of $n = 3 \pm$ SEM (biological replicates). Statistical analysis was performed using two-tailed Student's t test with Welch's correction. ***$p < 0.001$ ($p < 0.0001$ mock vs amlexanox (G), $p = 0.001$ mock vs H-151 (G), $p = 0.0001$ mock vs amlexanox (H), $p = 0.0002$ mock vs H-151 (H)). Source data are available online for this figure.

et al, 2022), potentially impairing protein quality control and promoting ER stress.

Even though the contribution of IFN-mediated inflammation to the overall phenotype is currently unclear, early anti-inflammatory treatment initiation represents a possible strategy to minimise irreversible organ damage (Rice et al, 2017). While conventional immunosuppressive agents such as corticosteroids appear to have limited efficacy in this context (Volpi et al, 2016; d'Angelo et al, 2021), JAK inhibition has shown some promise as a therapeutic option (Briand et al, 2019; Frémond et al, 2016, 2023; Sanchez et al, 2018; Meesilpavikkai et al, 2019; Vanderver et al, 2020). Presented here as a proof-of-principle, treatment of patient-derived fibroblasts with the TBK1 inhibitor amlexanox (Reilly et al, 2013; Bai et al, 2020), and the STING inhibitor H-151 (Tian et al, 2022), significantly decreased IFN signalling with relatively minimal cytotoxicity (Figs. 5G,H and EV5D,E).

In summary, our data dissects the contribution of ARF1-mediated ER-Golgi transport and maintenance of mitochondrial integrity for cGAS-STING signalling, showing that GTPase activity of ARF1 is required for trafficking, but dispensable for mitochondrial maintenance. Additionally, in vitro proof-of-concept treatment strategies selectively targeting the cGAS-STING axis might be relevant in reducing ARF1-related auto-inflammation.

## Methods

### Reagents and tools table

| Reagent/Resource | Reference or Source | Identifier or Catalog Number |
|---|---|---|
| **Experimental models** | | |
| 293-Dual-hSTING-R232 (*H. sapiens*) | InvivoGen | 293d-r232 |
| HDF (isolated from patient AGS3238) | This study | N/A |
| HDF (isolated from patient AGS460) | Hirschenberger et al (2023) | N/A |
| HDF (*H. sapiens*) | Innoprot | P10856 |
| HDF (*H. sapiens*) | Thermo Fisher Scientific | C0045C |
| HDF (*H. sapiens*) | Promocell | C-12300 |
| HEK293T (*H. sapiens*) | ATCC | CRL-3216, RRID: CVCL_0063 |
| Hela (*H. sapiens*) | ATCC | CCL-2, RRID: CVCL_0030 |
| **Recombinant DNA** | | |
| cGAS 3x-FLAG | Jae U. Jung | N/A |
| pCMV6 | Michaela Gack | N/A |
| pCMV6-hARF1-myc-FLAG | Origene | PS100012 |
| pCMV6-hARF1-Q71L-myc-FLAG | Hirschenberger et al (2023) | N/A |
| pCMV6-hARF1-R19C-myc-FLAG | This study | N/A |
| pCMV6-hARF1-R99C-myc-FLAG | Hirschenberger et al (2023) | N/A |
| pEGFP-C3-STING-WT | Hirschenberger et al (2023) | N/A |
| pEGFP-VSVGtsO45-KDELR2 | Hirschenberger et al (2023) | N/A |
| pGAPDH_PROM_01_Renilla SP Luciferase | Switchgear genomics | S721624 |
| pISRE-Fluc | Agilent | 219092 |
| pIRES-STING-FLAG | Jae U. Jung | N/A |
| **Antibodies** | | |
| Mouse anti-FLAG M2 | Sigma-Aldrich | F1804 |
| Rabbit anti-ARF1 | Proteintech | 20226-1-AP |
| Rabbit anti-COPG | Proteintech | 12393-1-AP |

| Reagent/Resource | Reference or Source | Identifier or Catalog Number |
|---|---|---|
| Rabbit anti-cGAS | Proteintech | 26416-1-AP |
| Rabbit anti-ERGIC-53 | Proteintech | 13364-1-AP |
| Rabbit anti-GM130 | Cell Signaling | 12480 |
| Rabbit anti-LC3 | Sigma-Aldrich | L8918 |
| Rabbit anti-pTBK1 | Cell Signaling | 5483 |
| Rabbit anti-STING | Proteintech | 19851-1-AP |
| Rabbit anti-TBK1 | Cell Signaling | 3504 |
| Rabbit anti-TFAM | Proteintech | 22586-1-AP |
| Rat anti-GAPDH | BioLegend | 607902 |
| Sheep anti-STING | Bio-Techne | AF6516 |
| Donkey anti-mouse IgG (H + L) Alexa Fluor Plus 568 | Thermo Fisher | A10037 |
| Donkey anti-rabbit IgG (H + L) Alexa Fluor Plus 568 | Thermo Fisher | A10042 |
| Donkey anti-rabbit IgG (H + L) Alexa Fluor Plus 647 | Thermo Fisher | A32795 |
| Donkey anti-sheep IgG (H + L) Alexa Fluor Plus 647 | Thermo Fisher | A-21448 |
| IRDye 680RD Goat anti-Mouse IgG Secondary Antibody | LI-COR | 926-68070 |
| IRDye 680RD Goat anti-Rabbit IgG Secondary Antibody | LI-COR | 926-68071 |
| IRDye 680RD Goat anti-Rat IgG Secondary Antibody | LI-COR | 926-68076 |
| IRDye 800CW Goat anti-Mouse IgG Secondary Antibody | LI-COR | 926-32210 |
| IRDye 800CW Goat anti-Rabbit IgG Secondary Antibody | LI-COR | 926-32211 |
| IRDye 800CW Goat anti-Rat IgG Secondary Antibody | LI-COR | 926-32219 |
| **Oligonucleotides and other sequence-based reagents** | | |
| PCR primer | This study | Table 1 |
| SYBR Green qPCR primer | This study | Table 2 |
| TaqMan Gene Expression Assay GAPDH | Thermo Fisher | 4310884E |
| TaqMan Gene Expression Assay MX1 | Thermo Fisher | Hs00895608_m1 |
| TaqMan Gene Expression Assay OAS1 | Thermo Fisher | Hs00973637_m1 |
| TaqMan Gene Expression Assay RSAD2 | Thermo Fisher | Hs00369813_m1 |
| **Chemicals, Enzymes and other reagents** | | |
| 10X Bolt Sample Reducing Agent | Invitrogen | B0009 |
| 2'3'-cGAMP | InvivoGen | tlrl-nacga23 |
| 4X Bolt LDS Sample Buffer | Invitrogen | B0007 |
| 4X Protein Sample Loading Buffer | LI-COR | 928-40004 |

| Reagent/Resource | Reference or Source | Identifier or Catalog Number |
|---|---|---|
| ABT-737 | SYNkinase | 1001 |
| Alkaline Phosphatase Blue Microwell Substrate | Sigma-Aldrich | AB0100 |
| Amlexanox | Cayman Chemical | 14181 |
| Anti-FLAG M2 Magnetic Beads | Merck | M8823 |
| Bafilomycin A1 | Santa Cruz Biotechnology | sc-201550 |
| Blocker Casein in PBS | Thermo Scientific | 37528 |
| Carbonylcyanid-4-(trifluormethoxy) phenylhydrazon (FCCP) | Sigma-Aldrich | C2920 |
| CellTiter-Glo Luminescent Cell Viability Assay | Promega | G7570 |
| Chameleon Duo Pre-stained Protein Ladder | LI-COR | 928-60000 |
| Chloroform:Isoamyl alcohol 24:1 | SERVA | 39554.02 |
| Coelenterazine (native-CTZ) | PFK Biotech | 102173 |
| CutSmart Buffer | New England Biolabs | B7204S |
| DAPI (4′,6-Diamidino-2-Phenylindole, Dihydrochloride) | Invitrogen | D1306 |
| Dimethyl sulfoxide (DMSO) | Sigma-Aldrich | 1029521000 |
| DMEM | Gibco | 41965039 |
| Dual-Glo Luciferase Assay System | Promega | E2980 |
| Epoxy embedding medium | Sigma-Aldrich | 45345 |
| ethylenediaminetetraacetic acid (EDTA) | Sigma-Aldrich | E9884 |
| Fetal Bovine Serum, qualified | Gibco | 10270106 |
| Glycerol | Sigma-Aldrich | G5516 |
| Glycogen from oyster, solution | SERVA | 39766 |
| GTPase-Glo Assay | Promega | V7681 |
| H-151 | InvivoGen | inh-h151 |
| HEPES | Sigma-Aldrich | H3375 |
| Immobilon-FL PVDF membrane | Millipore | 05317-10EA |
| Intercept (TBS) Blocking Buffer | LI-COR | 927-60001 |
| Interferon-β (human, recombinant) | PeproTech | 300-02BC |
| L-Glutamine 200 mM | PAN-Biotech | P04-80050 |
| MES-SDS running buffer (20X) | Alfa Aesar | J62138.K2 |
| Methylthiazolyldiphenyl-tetrazolium bromide (MTT) | Sigma-Aldrich | M5655 |
| Mitotracker green | Thermo Scientific | M7514 |

| Reagent/Resource | Reference or Source | Identifier or Catalog Number |
|---|---|---|
| MitoTracker Red CMXRos | Thermo Scientific | M7512 |
| Mowiol 4-88 | Carl Roth | 0713 |
| NuPAGE 4 to 12% Bis-Tris gel | Invitrogen | NP0321BOX |
| Opti-MEM | Gibco | 31985070 |
| Orange G | Sigma-Aldrich | O3756 |
| Osmium tetroxide | Sigma-Aldrich | 201030 |
| Paraformaldehyde solution 4% in PBS | Santa Cruz Biotechnology | sc-281692 |
| Passive Lysis 5X Buffer | Promega | E1941 |
| Penicillin-Streptomycin, 10,000 U/ml Penicillin, 10 mg/ml Streptomycin | PAN Biotech | P06-07100 |
| Phenol/Chloroform/Isoamyl alcohol | Carl Roth | A156.3 |
| Phosphate-Buffered Saline (PBS) | Gibco | 14190144 |
| Pierce Rapid Gold BCA Protein Assay Kit | Thermo Scientific | A53225 |
| Polyethylenimine, branched | Sigma-Aldrich | 408727 |
| PowerUp SYBR Green Master Mix | Applied Biosystems | A25742 |
| Protease Inhibitor Cocktail powder | Sigma-Aldrich | P2714 |
| Q5 Site-Directed Mutagenesis Kit | New England BioLabs | E0554S |
| Quick-RNA Microprep Kit | Zymo Research | R1051 |
| Q-VD-OPH | Cayman Chemical | 15260 |
| Saponin | Sigma-Aldrich | 47036 |
| Semi dry blot transfer buffer (10X) | Alfa Aesar | J63664.K3 |
| Sodium Dodecyl Sulfate (SDS) | Sigma-Aldrich | 822050 |
| SuperScript III Platinum One-Step qRT-PCR Kit | Thermo Fisher | 11732088 |
| TaqMan Gene Expression Master Mix | Applied Biosystems | 4369016 |
| Torin-1 | EZSolution | 2353 |
| TransIT-LT1 Transfection Reagent | Mirus | MIR 2300 |
| Triethylenediamine (DABCO) | Carl Roth | 0718 |
| Tris | AppliChem GmbH | A2264 |
| Triton X-100 | Sigma-Aldrich | T8787 |
| Trypsin 0.05%/EDTA 0.02% in PBS, w/o: Ca and Mg | PAN Biotech | P10-023100 |
| Tween 20 | Sigma-Aldrich | P9416 |
| UltraPure DNase/RNase-Free Distilled Water | Invitrogen | 10977015 |

| Reagent/Resource | Reference or Source | Identifier or Catalog Number |
|---|---|---|
| Uranyl Acetate Solution | Science Services | E22400 |
| Whatman filter paper | VWR | 588-3148 |
| Zenon Alexa Fluor 488 mouse IgG1 labelling reagent | Thermo Fisher | Z25302 |
| Zenon Alexa Fluor 568 rabbit IgG labelling reagent | Thermo Fisher | Z25306 |
| Zenon Alexa Fluor 647 rabbit IgG labelling reagent | Thermo Fisher | Z25308 |
| β-mercaptoethanol | Sigma-Aldrich | 444203 |
| **Software** | | |
| Corel Draw 2021 | Corel Corporation | https://www.coreldraw.com |
| GraphPad Prism 10 | GraphPad Software | https://www.graphpad.com |
| Huygens Professional 19.10 | Scientific Volume Imaging | ttps://www.svi.nl |
| ImageJ (Fiji) | N/A | https://imagej.net/Fiji |
| LI-COR Image Studio Lite Version 5.0.21 | LI-COR | https://www.licor.com |
| LI-COR Image Studio Version 5.2 | LI-COR | https://www.licor.com |
| Microsoft Office Professional Plus 2019 | Microsoft | https://www.microsoft.com |
| Simplicity 4.20 | Berthold Technologies GmbH & Co. KG | https://www.berthold.com/en/ |
| SoftMax Pro 7 | Molecular Devices LLC | https://www.moleculardevices.com |
| UCSF Chimera 1.15 | RBVI | https://www.cgl.ucsf.edu/chimera/ |
| ZEN 2010 | Zeiss | https://www.zeiss.com/ |
| **Other** | | |
| EM AFS2 | Leica Microsystems GmbH | N/A |
| HPF Compact 01 | Technotrade International Inc | N/A |
| Jeol JEM-1400 | Jeol Ltd | N/A |
| LI-COR Odyssey | LI-COR | N/A |
| Orion II microplate Luminometer | Berthold | N/A |
| StepOnePlus Real-Time PCR System | Applied Biosystems | N/A |
| Ultracut UC7 | Leica Microsystems GmbH | N/A |
| Veleta CCD | Olympus Life Science | N/A |
| Vmax kinetic microplate reader | Molecular Devices LLC | N/A |
| Zeiss Axiovert 40 CFL | Zeiss | N/A |
| Zeiss LSM 710 | Zeiss | N/A |

## Methods and protocols

### Cell culture

HEK293T (ATCC, CRL-3216), Hela (ATCC, CCL-2), and 293-Dual-hSTING-R232 (Invivogen, 293d-r232) cell lines, as well as normal human dermal fibroblasts (Thermo Fisher, C0045C; Innoprot, P10856; Promocell, C-12300), were maintained in Dulbecco's modified Eagle medium (DMEM) supplemented with 10% fetal bovine serum (FBS), 6.5 µg/ml gentamicin, and 2 mM L-glutamine. Cells isolated from patients AGS460 or AGS3238 via skin punch biopsies were cultivated under the same conditions. All cultures were kept at 37 °C in an atmosphere of 5% $CO_2$ and 90% humidity and are regularly checked for the presence of mycoplasma.

### Expression constructs and cloning

The plasmid encoding human ARF1 (pCMV6-hARF1-myc-FLAG) was acquired from Origene (PS100012, provided by Michaela Gack, Florida) and mutations at R99C, Q71L and R19C were introduced by Q5 site-specific mutagenesis (see Table 1 for primer sequences). Constructs coding for cGAS 3×-FLAG and STING-FLAG were kindly provided by Jae U. Jung (University of Southern California). pEGFP-C3-STING-WT and pEGFP-VSVGtsO45-KDELR2 were described previously (Hirschenberger et al, 2023). The plasmids pGAPDH_PROM_01_Renilla SP Luciferase (SwitchGear Genomics, Cat#S721624) and pISRE-FLuc (Stratagene, Agilent, Cat#219092) were purchased from the manufacturers.

### Transient transfection of mammalian cells

Transient transfection of plasmid DNA was performed utilizing either the TransIT-LT1 Transfection Reagent (Mirus, MIR 2300) or Polyethylenimine (PEI, 1 mg/ml in $H_2O$), adhering to the manufacturers' instructions or as previously described (Koepke et al, 2020).

### SDS-PAGE and immunoblotting

SDS-PAGE and immunoblotting were executed following standard protocols (Koepke et al, 2020). Initially, cell lysates were mixed with 6x Protein Sample Loading Buffer (1x final dilution) containing 15% β-mercaptoethanol and heated to 95 °C for 10 min. Subsequently, separation was performed on NuPAGE 4–12% Bis-Tris Gels (Invitrogen, NP0321BOX) for 90 min at 90 V, followed by transfer onto Immobilon-FL PVDF membranes (Merck Millipore, 05317-10EA) at a constant voltage of 30 V for 30 min. Post-transfer, the membrane underwent blocking in 1% Casein in PBS. The primary antibodies mouse anti-FLAG M2 (1:5000, Sigma-Aldrich, F1804), sheep anti-STING (1:1000, Bio-Techne, AF6516), rabbit anti-STING (1:1000, Proteintech, 19851-1-AP), rabbit anti-TFAM (1:1000, Proteintech, 22586-1-AP), rabbit-anti-COPG (1:1000, Proteintech, 12393-1-AP)), rabbit anti-pTBK1 (1:1000, Cell Signaling, 5483), rabbit anti-TBK1 (1:1000, Cell Signaling, 3504), rabbit anti-LC3 (1:1000, Sigma-Aldrich, L8918), rabbit anti-ARF1 (1:500, Proteintech, 20226-1-AP) and rat anti-GAPDH (1:1000, BioLegend, 607902) were used for protein staining. Subsequent detection utilized infrared Dye labelled secondary antibodies (LI-COR, IRDye 680RD Goat anti-Mouse IgG Secondary Antibody, 926-68070; IRDye 680RD Goat anti-Rabbit IgG Secondary Antibody, 926-68071; IRDye 680RD Goat anti-Rat IgG Secondary Antibody, 926-68076; IRDye 800CW Goat anti-Mouse IgG Secondary Antibody, 926-32210; 926-32211; IRDye 800CW Goat anti-Rat IgG Secondary Antibody, 926-32219) diluted in 0.05% Casein in PBS. Image Studio lite (LI-COR) was utilized for quantification of band intensities.

### Reporter gene assays

To analyse the impact of proteins of interest on ISRE induction, HEK293T cells or 293-Dual hSTING-R232 reporter cells were used. HEK293T cells were transfected with an ISRE-inducible Firefly luciferase (Fluc) reporter and a GAPDH-promoter controlled Renilla luciferase (Rluc) reporter, along with the constructs of interest, utilizing TransIT-LT1 transfection (Hirschenberger et al, 2021). 293-Dual hSTING-R232 cells, which stably express an ISRE-inducible secreted embryonic alkaline phosphatase (SEAP) reporter construct and a secreted Lucia luciferase regulated by the endogenous IFN-ß promoter, were transiently transfected using TransIT-LT1 transfection or stimulated with fibroblast supernatant. Cells were either harvested 37–40 h post-transfection in passive lysis buffer (Promega, E1941), or the supernatant of the 293-Dual hSTING-R232 cells was collected 48–72 h post SN transfer. ISRE-Fluc reporter activity was quantified using the Promega luciferase assay system (Promega, E1501) and GAPDH-Rluc activity was measured utilizing the Promega Renilla luciferase assay system (Promega, E2820), following the manufacturer's instructions. Measurements were performed with the Orion II microplate luminometer and Simplicity software (Berthold), with ISRE-Fluc values normalized to GAPDH-Rluc values. ISRE-SEAP activity was analysed via the addition of the alkaline phosphatase blue microwell substrate (Sigma-Aldrich, AB0100), and quantified with the Vmax kinetic microplate reader and SoftMax Pro 7 software (Molecular Devices) at specified wavelengths. ISRE-SEAP values were then normalized to viable cell numbers determined by the CellTiter-Glo luminescent cell viability assay (Promega, G7570).

### Supernatant transfer

In supernatant (SN) transfer experiments, cell culture media from primary human dermal fibroblasts (healthy donors, patient AGS3238 or patient AGS460), following the indicated treatments, were transferred to 293-Dual-hSTING-R232 reporter cells. ISRE-SEAP activity was assessed 48–72 h post-SN transfer, employing the standard reporter gene assay procedure.

### Cell fractionation and quantification of mitochondrial DNA

HEK293T cells were transfected with either ARF1 WT, ARF1 R99C, ARF1 R19C, or an empty vector. After 24 h, the positive control cells were treated with 10 µM ABT-737 (SYNkinase, 1001) and 10 µM Quinoline-Val-Asp-Difluorophenoxymethylketone (Q-VD-OPH, Cayman Chemical, 15260). The next day, cell harvesting and DNA isolation and quantification from cytosolic and mitochondrial fractions were performed as previously described (basic protocol 2) (Hirschenberger et al, 2023; Bryant et al, 2022). In detail, half of the cells were lysed in SDS lysis buffer (20 mM Tris, pH 8, 1% SDS, protease inhibitors) to prepare WCLs for normalization, while the remaining cells were fractionated. Cytosolic and mitochondrial extracts were obtained by sequential treatment with saponin lysis buffer (1x PBS, pH 7.4, 0.05% saponin, protease inhibitors) and NP-40 lysis buffer (50 mM Tris, pH 7.5, 150 mM NaCl, 1 mM EDTA, 1% NP-40, 10% glycerol, protease

inhibitors), respectively. The purity of the extracts was confirmed by immunoblotting for GAPDH (cytosolic extract) and TFAM (mitochondrial extract). DNA from the fractions and WCLs was extracted using phenol-chloroform, and DNA concentrations were measured using a Nanodrop photometer. Equal amounts of DNA were used for qPCR quantification of mitochondrial DNA (MT-D-L.oop, MT-ND1) and nuclear DNA (KCNJ10) (primer sequences in Table 2) using the PowerUP SYBR Green Kit (Applied Biosystems, A25742), and relative cytosolic mtDNA levels were calculated using the ΔΔCT method.

### Mitochondrial membrane potential (MMP) measurements

HEK293T cells were seeded and transfected the following day with either ARF1 WT, ARF1 R19C, or an empty vector using PEI. Treatment with FCCP (10 μM/4 h, Sigma-Aldrich, C2920) was used as control. 48 h after transfection, cells were harvested by trypsinization and transferred to FACS tubes. After centrifugation at 1800 rpm for 5 min, cells were washed with cold PBS and centrifuged again to collect cells. Supernatant was discarded and cells were stained with Mitotracker green (500 nM, Thermo Fisher Scientific, M7514) and MitoTracker Red CMXRos (100 nM, Thermo Fisher Scientific, M7512) for 30 min at 37 °C. After 30 min, cells were washed with PBS and centrifuged again for 5 min at 1800 rpm. Supernatant was discarded, samples were resuspended in PBS and acquired at the Attune NxT Flow Cytometer (Thermo Fisher Scientific).

### Immunofluorescence microscopy

Cells were cultured on coverslips placed in 24-well plates and subjected to the specified transfections or treatments. Subsequently, the samples were rinsed with PBS and fixed with a 4% paraformaldehyde solution for 20 min at RT. For permeabilization and blocking, the cells were incubated in PBS with 0.5% Triton X-100 and 5% FCS for 1 h at RT. Following this, cells were washed again with PBS and incubated for 2 h at 4 °C with the primary antibodies mouse anti-FLAG M2 (1:400, Sigma-Aldrich, F1804), rabbit anti-GM130 (1:400, Cell Signaling, 12480), rabbit anti-ERGIC-53 (1:400, Proteintech, 13364-1-AP), rabbit anti-pTBK1 (1:100, Cell Signaling, 5483) or rabbit anti-STING (1:200, Proteintech, 19851-1-AP), all diluted in PBS containing 1% FCS. The samples were then washed with PBS/0.1% Tween 20 and incubated with secondary antibodies donkey anti-mouse IgG (H + L) Alexa Fluor Plus 568 (A10037), donkey anti-rabbit IgG (H + L) Alexa Fluor 568 (A10042), donkey anti-rabbit IgG (H + L) Alexa Fluor Plus 647 (A32795) (all 1:400, Thermo Fisher Scientific) or with primary antibody-secondary antibody conjugates (rabbit anti-ERGIC-53, 1.25 μg/ml; rabbit anti-GM130, 0.585 μg/ml; rabbit anti-pTBK1, 1.42 μg/ml; rabbit anti-STING, 3.5 μg/ml; conjugated to equal amounts (μg/ml) of Zenon Alexa Fluor 647 rabbit IgG labelling reagent (Z25308), Zenon Alexa Fluor 568 rabbit IgG labelling reagent (Z25306), Zenon Alexa Fluor 488 rabbit IgG labelling reagent (Z25002) (Thermo Fisher Scientific)) and 500 ng/ml DAPI (Invitrogen, D1306) for 2 h at 4 °C in the dark. The samples were then washed with PBS/0.1% Tween 20 and water, and the coverslips were mounted on microscopy slides using Mowiol mounting medium (10% (w/v) Mowiol 4–88, 25% (w/v) Glycerol, 50% (v/v) 0.2 mM Tris-HCl (pre-adjusted to pH 8.5) and 2.5% (w/v) DABCO dissolved in water). Imaging was performed with a Zeiss LSM 710 confocal laser scanning microscope using the ZEN 2010 software, and image analysis was conducted with ImageJ (Fiji). Colocalization was assessed using the

Huygens Professional 19.04 software, calculating Pearson coefficients with the "Huygens Colocalization Analyzer" using the Costes method (Costes et al, 2004) and individual thresholds. For analysing pTBK1 particles, ImageJ (Fiji) software was utilized, setting thresholds based on the negative control background, and using the commands "Despeckle", "Analyze Particles …", and "exclude clear summarize" to determine the percentage of the cell area occupied by pTBK1 punctae.

### VSV-G retrograde transport assay

To study retrograde Golgi to ER trafficking, a temperature-sensitive VSVGtsO45 fusion protein was used as a model cargo as described previously (Hirschenberger et al, 2023; Cole et al, 1998). In short, HeLa cells were transfected with pEGFP-VSVG-ts045-KDELR along with either ARF1 WT, ARF1 R19C, ARF1 R99C, or an empty vector. The cells were maintained at 37 °C for 24 h after which they were either fixed immediately or incubated at 32 °C for 2 h to accumulate the fusion protein at the Golgi. Subsequently, the cells were either fixed at this stage or shifted to 40 °C for 1 h to facilitate a round of retrograde transport from the Golgi to the ER before being fixed.

### In vitro GTPase assay

HEK293T cells were transfected with ARF1 WT, ARF1 R19C, ARF1 R99C, ARF1 Q71L, or an empty vector. After 24 h, WCLs were prepared using GTPase lysis buffer (150 mM NaCl, 50 mM HEPES, pH 7.4, 1% Triton X-100, 5 mM MgCl$_2$, and 5 mM EDTA). The cell lysates were vortexed for 30 s, incubated on ice for 10 min, and then centrifuged at $15,000 \times g$ for 20 min at 4 °C. The protein concentrations of the SNs were adjusted with the Pierce Rapid Gold bicinchoninic acid (BCA) Protein Assay Kit according to the manufacturer's instructions. The samples were then incubated with anti-FLAG M2 magnetic beads (Sigma-Aldrich, M8823) for 4 h at 4 °C on a rotating shaker. Following incubation, the beads were washed three times with washing buffer I (500 mM NaCl, 50 mM HEPES, pH 7.4, 1% Triton X-100, 5 mM MgCl$_2$, 5 mM EDTA) and twice with washing buffer II (100 mM NaCl, 50 mM HEPES, pH 7.4, 1% Triton X-100, 5 mM MgCl$_2$, 5 mM EDTA). The GTPase reaction was then performed as per the manufacturer's recommendations using the GTPase-Glo assay (Promega, V7681). In brief, the beads were resuspended in GEF Buffer, and 2x GTP solution was added to initiate the GTPase reaction. After a 12-h incubation at 37 °C, GTPase activity was quantified by adding the GTPase-Glo reagent and the detection reagent to the supernatant. The resulting luminescence was measured using an Orion II microplate Luminometer with the Simplicity software (Berthold).

### Co-immunoprecipitation

HEK293T cells were transfected with either FLAG-tagged ARF1 WT, ARF1 R19C, ARF1 R99C, or a vector control. For immunoprecipitation of endogenous ARF1 HDFs from patient AGS3238 or a healthy control donor were used. 24 h after transfection (HEK293T) or seeding (HDFs), WCLs were prepared by collecting the cells in PBS (Gibco, 14190144). Cell pellets were obtained by centrifugation ($500 \times g$, 5 min, 4 °C) and then lysed in transmembrane lysis buffer (150 mM NaCl, 50 mM HEPES, pH 7.4, 1% Triton X-100, 5 mM EDTA) by vortexing at maximum intensity for 30 s. Cell debris was removed by centrifugation ($20,000 \times g$, 20 min, 4 °C). The protein concentration in the supernatants was

**Table 1. Primers used for cloning.**

| Name | Sequence 5'–3' |
| --- | --- |
| ARF1-Q71L-fwd | TTG GAC AAG ATC CGG CCC CTG TGG |
| ARF1-Q71L-rev | GCC ACC CAC GTC CCA CAC AGT |
| ARF1-R19C-fwd | CAA AAA AGA AAT GTG CAT CCT CAT GG |
| ARF1-R19C-rev | CCA AAA AGG CCC TTG AAG AG |
| ARF1 R99C-fwd | GAG TGT GTG AAC GAG GCC CGT GAG GAG C |
| ARF1 R99C-rev | TCT GTC ATT GCT GTC CAC CAC GAA GAT C |

**Table 2. Primers used for SYBR Green qPCR.**

| PCR Type | Name | Sequence 5'–3' |
| --- | --- | --- |
| SYBR Green | KCNJ10 fwd | GCGCAAAAGCCTCCTCATT |
| SYBR Green | KCNJ10 rev | CCTTCCTTGGTTTGGTGGG |
| SYBR Green | MT-D-Loop fwd | CATAAAGCCTAAATAGCCCACACG |
| SYBR Green | MT-D-Loop rev | CCGTGAGTGGTTAATAGGGTGATA |
| SYBR Green | MT-ND1 fwd | GAACTAGTCTCAGGCTTCAACATCG |
| SYBR Green | MT-ND1 rec | CTAGGAAGATTGTAGTGGTGAGGGTG |

determined using a BCA assay (Pierce Rapid Gold BCA Protein Assay Kit, Thermo Fisher Scientific, A53225), and aliquots were saved for western blotting. The WCLs were then incubated with anti-FLAG (HEK293T) or anti-ARF1 (HDFs)-coupled dynabeads (Invitrogen, 10004D) for 4 h at 4 °C with continuous rotation. The beads were subsequently washed five times with transmembrane lysis buffer containing 300 mM NaCl and then incubated with 1x Protein Sample Loading Buffer supplemented with 15% β-mercaptoethanol. Following a 10-min incubation at 95 °C, the samples were analysed by SDS-PAGE and immunoblotting.

### Electron microscopy

EM sample preparation was carried out as described previously (Bergner et al, 2022). Primary human dermal fibroblasts (healthy donor, patient AGS3238, patient AGS460) or HEK293T cells were cultivated on carbon-coated and UV-sterilized sapphire disks (diameter: 3 μm, thickness: 160 μm Engineering Office M.). HEK293T cells were transfected with ARF1 WT, ARF1 R19C, ARF1 R99C, or an empty vector. After 24 h, the samples were cryo-immobilised in a Wohlwend HPF compact 01 high-pressure freezer. Subsequently, the samples were freeze-substituted in a solution containing 0.1% uranyl acetate (UA), 0.2% osmium tetroxide (OsO4) in acetone and 5% double-distilled water to enhance membrane visibility (Walther and Ziegler, 2002). Freeze substitution was performed in an EM AFS2 (Leica Microsystems GmbH) device, where the temperature was gradually increased from −90 to 0 °C over a 17 h period, with a one-hour incubation step at 0 °C and a further increase to RT for 1 h. Subsequently, the samples were washed three times with acetone for 30 min at RT and incrementally infiltrated with EPON resin (30%, 60% and 100% EPON in acetone, Sigma-Aldrich, 45345). Polymerization of EPON was finally initiated by incubating the samples at 60 °C for 48 h. The sapphire disks were removed, leaving the cells on the surface of the EPON block. For transmission electron microscopy (TEM) analysis, 70 nm thin section were cut from the EPON block using a 45° diamond knife (Diatome Ltd) mounted on

an Ultracut UCT ultramicrotome (Leica) and collected on freshly glow-discharged, single slot copper grids coated with a carbon-reinforced Formvar film. The samples were examined with a JEM-1400 TEM (Jeol) operating at 120 kV acceleration voltage and images were acquired with a CCD camera (Veleta, Olympus Life Science). For improved visibility, lumina of stacked Golgi cisternae were manually highlighted using CorelDraw.

### EM quantification of Golgi morphology

Fibroblasts from a healthy donor or patient AGS3238 were imaged by TEM. For each cell, one cross section was analysed. All those Golgis were imaged which were cross sectioned in a way that the lipid bilayer of the Golgi membrane was visible. These Golgis were then categorized into normal or fragmented Golgis: Normal Golgis show the typical Golgi ribbon morphology with at least two stacked, parallel cisternae, each of which showed a similar width. Golgis were considered as fragmented if the compact, stacked cisternae arrangement was lost and the Golgi appeared as irregularly scattered cisternae and vesicles which often appeared dilated. The number of normal and fragmented Golgis was counted and their percentage in relation to the total number of analysed Golgis was calculated.

### Assessing cell viability

Cell viability was evaluated using either the CellTiter-Glo luminescent assay or the MTT-based colorimetric assay. For the CellTiter-Glo assay, CellTiter-Glo buffer and substrate were added to the cells and incubated for 10 min at RT as per the manufacturer's instructions. Measurements were performed using the Orion II microplate reader and the Simplicity software. The MTT assay, adapted with minor changes from the literature (van Meerloo et al, 2011), involved adding MTT solution (0.5 mg/ml) to adherent cells after removing the medium, followed by a 3 h incubation at 37 °C. The resulting formazan crystals were dissolved in a 1:1 mixture of dimethyl sulfoxide (DMSO) and ethanol. Absorbance was measured at 490 nm with a baseline correction at 650 nm using the Vmax kinetic microplate reader and SoftMax Pro 7 software.

### RNA isolation and qRT-PCR

Total cellular RNA was isolated using the Quick-RNA Microprep Kit (Zymo Research, R1051), following the manufacturer's protocol. Reverse transcription and qRT-PCR were carried out in a single step using the SuperScript III Platinum Kit (Thermo Fisher Scientific, 11732088) on a StepOnePlus Real-Time PCR System (Applied Biosystems), as per the manufacturer's instructions. Pre-mixed TaqMan Gene Expression Assays (Thermo Fisher Scientific) for each target gene were added to the reaction (MX1, Hs00895608_m1; OAS1, Hs00973637_m1; RSAD2, Hs00369813_m1; Thermo Fisher Scientific) and expression levels of individual genes were quantified by normalizing to GAPDH cDNA levels using the ΔΔCT method.

### Analysis of ISG mRNA levels in cGAMP-stimulated HDFs over time

To evaluate ISG expression levels over time, HDFs from patient AGS3238 and healthy control donors, were seeded and stimulated with 20 μg/ml of cGAMP the next day, followed by incubation at 37 °C. Cells were harvested at four distinct time points (1, 2, 3, and 5 days post-stimulation) to capture different treatment durations. Total RNA was isolated from the cells and ISG mRNA levels quantified via qPCR.

### Analysis of ISG mRNA levels in Poly I:C-stimulated HDFs

To evaluate ISG expression levels over time, HDFs from patient AGS3238 and healthy control donors, were seeded and stimulated with 1 µg/ml of Poly I:C the next day, followed by incubation at 37 °C. Cells were harvested 1- and 5-day(s) post-stimulation to capture an early and a late treatment duration. Total RNA was isolated from the cells and ISG mRNA levels quantified via qPCR.

### Stimulation of HDFs for Western blotting

HDFs from patient AGS3238 or a healthy control donor were stimulated with 20 µg/ml of cGAMP for 1 h, 3 h or 6 h or left untreated. Subsequently, cells were harvested by trypsinization. After centrifugation at 1800 rpm for 5 min, WCLs were prepared by collecting the cells in Phosphate-Buffered Saline (PBS, Gibco, 14190144). Cell pellets were obtained by centrifugation ($500 \times g$, 5 min, 4 °C) and then lysed in transmembrane lysis buffer (150 mM NaCl, 50 mM HEPES, pH 7.4, 1% Triton X-100, 5 mM EDTA) by vortexing at maximum intensity for 30 s. Cell debris was removed by centrifugation ($20,000 \times g$, 20 min, 4 °C). The protein concentration in the supernatants was determined using a BCA assay (Pierce Rapid Gold BCA Protein Assay Kit, Thermo Fisher Scientific, A53225). Cell lysates were mixed with 6x Protein Sample Loading Buffer (1x final dilution) containing 15% β-mercaptoethanol, heated to 95 °C for 10 min and then analysed by SDS-PAGE and immunoblotting.

### CRISPR/Cas9 KO and analysis of ISG mRNA levels in HDFs

HDFs from patient AGS3238 or a healthy control donor were nucleofected with a complex of HiFi Cas9 Nuclease V3 (80 pmol, IDT, 1081059) and gRNA (300 pmol, IDT) using a STING/TMEM173-specific sgRNAs (5′-ATTTCAACGTGGCCCATG-3′, Lonza) or a non-targeting (NT) control sgRNA (5′-ACGGAGGC-TAAGCGTCGCAA-3′). The nucleofection was performed with the P4 Primary Cell 4D-Nucleofector X-Kit (Lonza, V4XP-4024) on an Amaxa 4D-Nucleofector (Lonza) using pulse code EO114. 96 h post nucleofection the cells were harvested for western blotting or stimulated with cGAMP (20 µg/ml), Poly I:C (1 µg/ml), or LPS (5 mg/ml) for 16 h. Total RNA was isolated from the cells and ISG mRNA levels quantified via qPCR.

### cGAS-STING pathway inhibitor treatment

To evaluate inhibitor functionality, 293-Dual-hSTING-R232 reporter cells were first stimulated with 20 µg/ml of cGAMP before inhibitor treatment. Amlexanox (diluted in DMSO) was administered one-hour post-stimulation, while H-151 (diluted in DMSO) was given 2 h post-stimulation. Both inhibitors were tested at increasing concentrations (H-151: 0.1 µM, 1 µM, 2 µM, 5 µM, 10 µM; amlexanox: 10 µg/ml, 20 µg/ml, 33 µg/ml, 40 µg/ml, 50 µg/ml) and effectiveness assessed using a reporter gene assay. To investigate the impact of cGAS-STING pathway inhibitors on patient-derived fibroblasts, HDFs from patients AGS3238 and AGS460, along with a healthy control donor, were exposed to 2 µM of the STING inhibitor H-151, or 33 µg/ml of the TBK1/IKKε inhibitor amlexanox, one day post seeding or left untreated. The inhibitors were administered sequentially every two days (on day 3, day 5, and day 7), totalling four treatments. Cells were harvested one day after the final treatment, and SNs were transferred to 293-Dual-hSTING-R232 reporter cells for a reporter gene assay. Metabolic cell viability was assessed using MTT assay.

### Conservation and structural analysis of ARF1

The conservation of ARF1 was analysed using ConSurf, based on a protein sequence from the Protein Data Bank (PDB: 2J59) (Ménétrey et al, 2007). The crystal structure of GTP-bound ARF1, also obtained from PDB: 2J59, was visualized as a ribbon diagram with UCSF Chimera 1.6.1, highlighting residue R19 as a stick model.

### Ethical statement

Clinical information and samples were collected with informed consent and permission to publish the research. Consent to participate and publish was obtained from patients, or from their parent(s) or guardian(s) in the case of minors. The study and the handling of ARF1 patient cells received approval from the Comité de Protection des Personnes (ID-RCB/EUDRACT: 2014-A01017-40) in France, the Leeds (East) Research Ethics Committee (REC reference: 10/H1307/2 IRAS project ID: 62971) in the UK and the Ethics Committee at Ulm University (Approval 530/21). The study design did not consider sex and/or gender as these factors were irrelevant to the study. Individual patient information is detailed in the Appendix.

### Statistical analysis

Statistical analyses were performed with GraphPad Prism 10, employing a two-tailed Student's t-test with Welch's correction to determine p-values. Data are presented as the mean ± SEM from a minimum of three biological replicates, unless otherwise specified. Significant differences are denoted by asterisks ($*p < 0.05$, $**p < 0.01$, $***p < 0.001$), while non-significant (ns) differences are not highlighted unless explicitly noted.

## Data availability

This study includes no data deposited in external repositories. Patient exome and genome sequencing data are not publicly available due to the possibility of compromising privacy. All primary human fibroblasts used in this study were donated and are therefore a limited resource. Availability is through the corresponding author subject to availability and completion of a material transfer agreement required to ensure patient privacy.

The source data of this paper are collected in the following database record: biostudies:S-SCDT-10_1038-S44319-025-00423-7.

## Peer review information

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

## Acknowledgements

We thank Daniela Krnavek, Martha Mayer, Kerstin Regensburger, Regina Burger, Jana-Romana Fischer and Birgit Ott for assistance and Prof. Alexander Kühne (Ulm University) for providing access to the Leica SP5 microscope (DFG project ID: 432000323). KMJS acknowledges funding by the German Federal Ministry of Education and Research (BMBF; IMMUNOMOD-01KI2014) and the German Research Foundation (DFG; CRC1279, SP 1600/9-1). KMJS and YJC are additionally supported by a DFG/Agence Nationale de la Recherche (ANR, France) joint grant (ANR-22-CE92-0012/SP1600/7-1; acronym InnateTraffic). AL acknowledges funding from Inserm and ANR (ANR 2022 – MITIgATe). YJC acknowledges the European Research Council (786142 E-T1IFNs), a UK Medical Research Council Human Genetics Unit core grant (MC_UU_00035/11), a state subsidy from the Agence Nationale de la Recherche (France) under the Investissements d'avenir programme bearing the reference ANR-10-IAHU-01. KS-K acknowledges funding by the German Research Foundation (DFG; CRC1506; Project-ID 450627322 and CRC1149; Project-ID 251293561). JZ is a fellow of the Margarete von Wrangell-Habilitationsprogramm (Ministry of Science, Research and Arts Baden-Wuerttemberg, European Social Fund) and receives funding from the DFG (project number: 520584003). We thank the ULMTeC Core Facilities of the Medical Faculty, Ulm University, (Electron Microscopy) for their support.

## Author contributions

**Johannes Lang**: Conceptualization; Data curation; Formal analysis; Investigation; Visualization; Methodology; Writing—review and editing. **Tim Bergner**: Formal analysis; Investigation; Visualization; Methodology; Writing—review and editing. **Julia Zinngrebe**: Data curation; Formal analysis; Funding acquisition; Investigation; Methodology; Writing—review and editing. **Alice Lepelley**: Conceptualization; Resources; Formal analysis; Supervision; Funding acquisition; Investigation; Methodology; Writing—review and editing. **Katharina Vill**: Resources; Investigation; Writing—review and editing. **Steffen Leiz**: Resources; Investigation; Writing—review and editing. **Meinhard Wlaschek**: Resources; Supervision; Investigation; Writing—review and editing. **Matias Wagner**: Resources; Investigation; Writing—review and editing. **Karin Scharffetter-Kochanek**: Supervision; Funding acquisition; Writing—review and editing. **Pamela Fischer-Posovszky**: Data curation; Formal analysis; Supervision; Investigation; Writing—review and editing. **Clarissa Read**: Data curation; Formal analysis; Supervision; Funding acquisition; Writing—review and editing. **Yanick J Crow**: Conceptualization; Formal analysis; Supervision; Funding acquisition; Writing—review and editing. **Maximilian Hirschenberger**: Conceptualization; Data curation; Formal analysis; Supervision; Investigation; Visualization; Methodology; Writing—original draft; Project administration; Writing—review and editing. **Konstantin M J Sparrer**: Conceptualization; Data curation; Formal analysis; Supervision; Funding acquisition; Visualization; Writing—original draft; Project administration; Writing—review and editing.

Source data underlying figure panels in this paper may have individual authorship assigned. Where available, figure panel/source data authorship is listed in the following database record: biostudies:S-SCDT-10_1038-S44319-025-00423-7.

## Funding

## Disclosure and competing interests statement

The authors declare no competing interests.

# Expanded View Figures

**Figure EV1. ARF1 expression and impact on mitochondrial integrity.**

(A) Representative immunoblots of HEK293T cells transiently expressing increasing amounts of FLAG-tagged ARF1 WT, R19C or R99C, together with STING (+STING) or vector control (-STING. Blots were stained with anti-GAPDH. (B) Representative immunoblots of HEK293T cells transiently expressing increasing amounts (10–100 ng, in steps of 10 ng) of FLAG-tagged ARF1 WT, R19C or R99C, together with STING. Blots were stained with anti-GAPDH. (C) qPCR analysis of mitochondrial (mt) DNA (MT-ND-1) in the cytosolic fraction of Fig. 2B, normalised to total cellular mtDNA (MT-ND-1/KCNJ10). Bars represent the mean of $n = 6 \pm$ SEM (biological replicates). Statistical analysis was performed using two-tailed Student's t test with Welch's correction. *$p < 0.05$ ($p = 0.0184$ WT vs ABT/QVD); ns, not significant ($p = 0.0921$ WT vs R19C, $p = 0.0542$ WT vs R99C)). (D) Mitochondrial membrane potential (MMP) of HEK293T cell expressing ARF1 WT, R19C, R99C or vector control or treated with FCCP (10 µM/ 4 h). MMP was measured using flow cytometry and data depicted as the ratio of Mitotracker Red MFI (MMP) to Mitotracker Green MFI (mitochondrial mass), expressed as percentage of the vector control. Lines represent the mean of $n = 3 \pm$ SEM (biological replicates). Statistical analysis was performed using two-tailed Student's t test with Welch's correction. ***$p < 0.001$ ($p < 0.0001$ WT vs FCCP); ns, not significant ($p = 0.9464$ WT vs R19C). (E) Representative immunoblots of HEK293-STING cells transiently expressing increasing amounts of FLAG-tagged ARF1 WT, R19C, R99C or vector control. Blots were stained with anti-STING and anti-GAPDH.

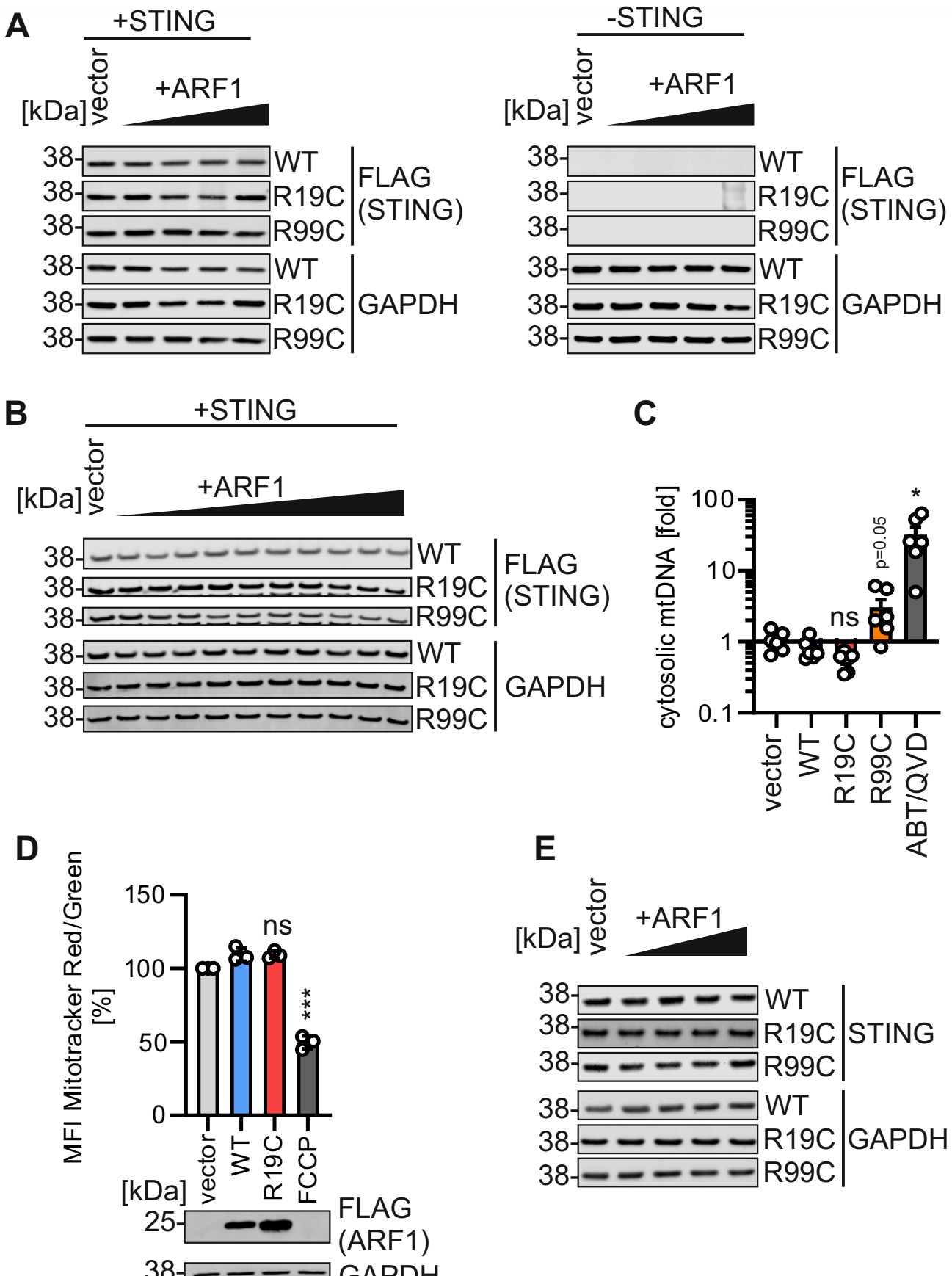

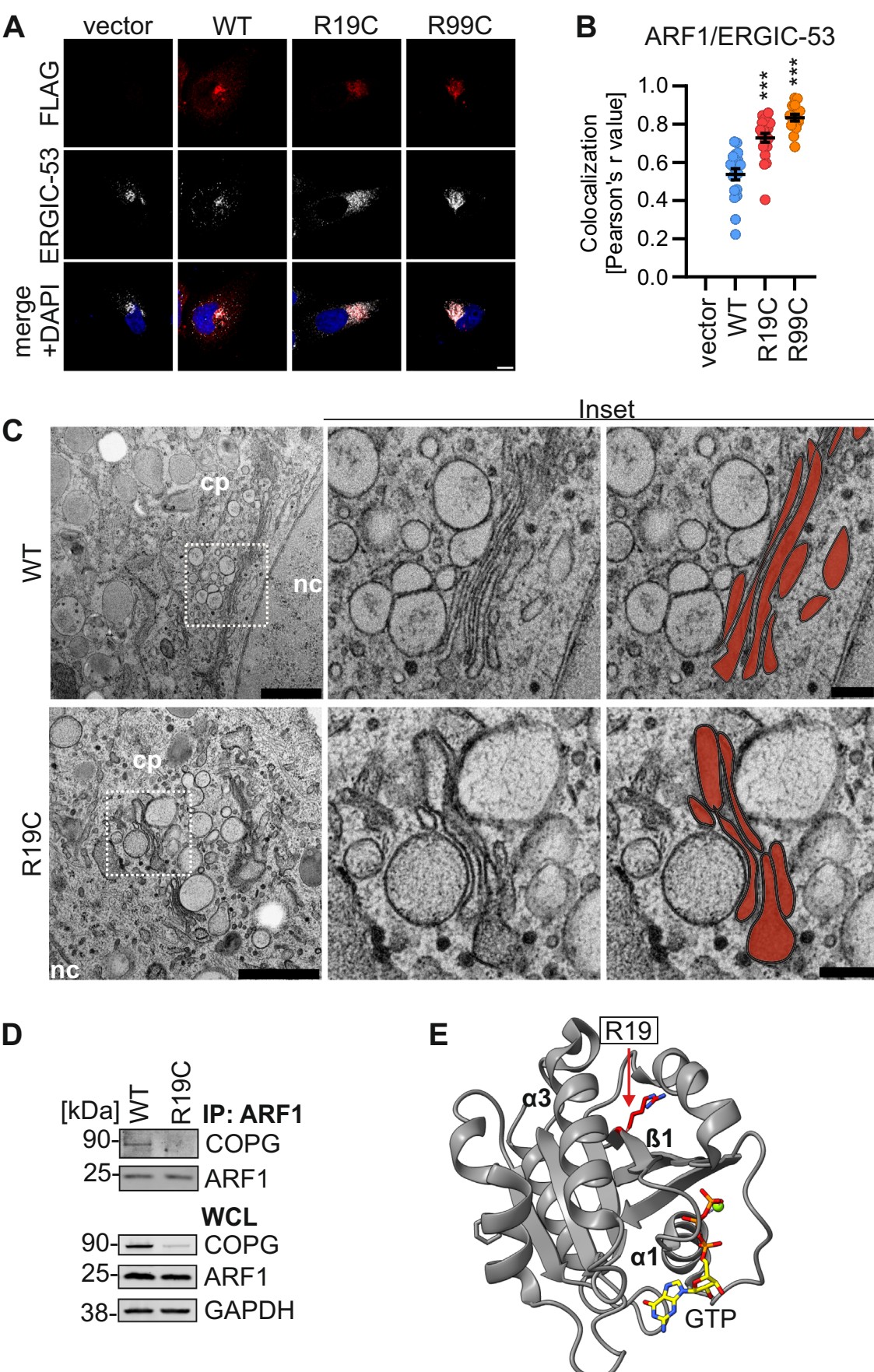

◀ **Figure EV2. Localisation of ARF1 R19C and the impact on Golgi morphology and COPG interaction.**

(A) Representative immunofluorescence images of Hela cells expressing ARF1 WT, R19C, R99C or vector control. 24 h post transfection the cells were stained with anti-FLAG (red) and anti- ERGIC-53 (white). Nuclei: DAPI (blue). Scale bar: 10 μm. (B) Colocalization (Pearson's correlation coefficient, r) between ARF1 and ERGIC-53 from the images shown in (A). Lines represent the mean of $n = 16$–$21 \pm$ SEM (individual cells). Statistical analysis was performed using two-tailed Student's t test with Welch's correction. ***$p < 0.001$ ($p < 0.0001$ WT vs R19C, WT vs R99C). (C) Representative transmission electron microscopy images of the Golgi area of primary fibroblasts from a healthy donor (WT) or patient AGS3238 (R19C). Stacked Golgi cisternae are highlighted in red. cp, cytoplasm. nc, nucleus. Scale bar: 1 μm (overview images), 200 nm (higher magnification image). (D) Immunoprecipitation (IP) of WCLs of primary fibroblasts from a healthy donor (WT) or patient AGS3238 (R19C) using anti-ARF1 coupled dynabeads. Immunoblots of the WCL and the IP were stained with anti-COPG, anti-ARF1 and anti-GAPDH. (E) Ribbon diagram of ARF1 in GTP-bound state (PDB: 2J59). R19 is highlighted in red, GTP in yellow. Source data are available online for this figure.

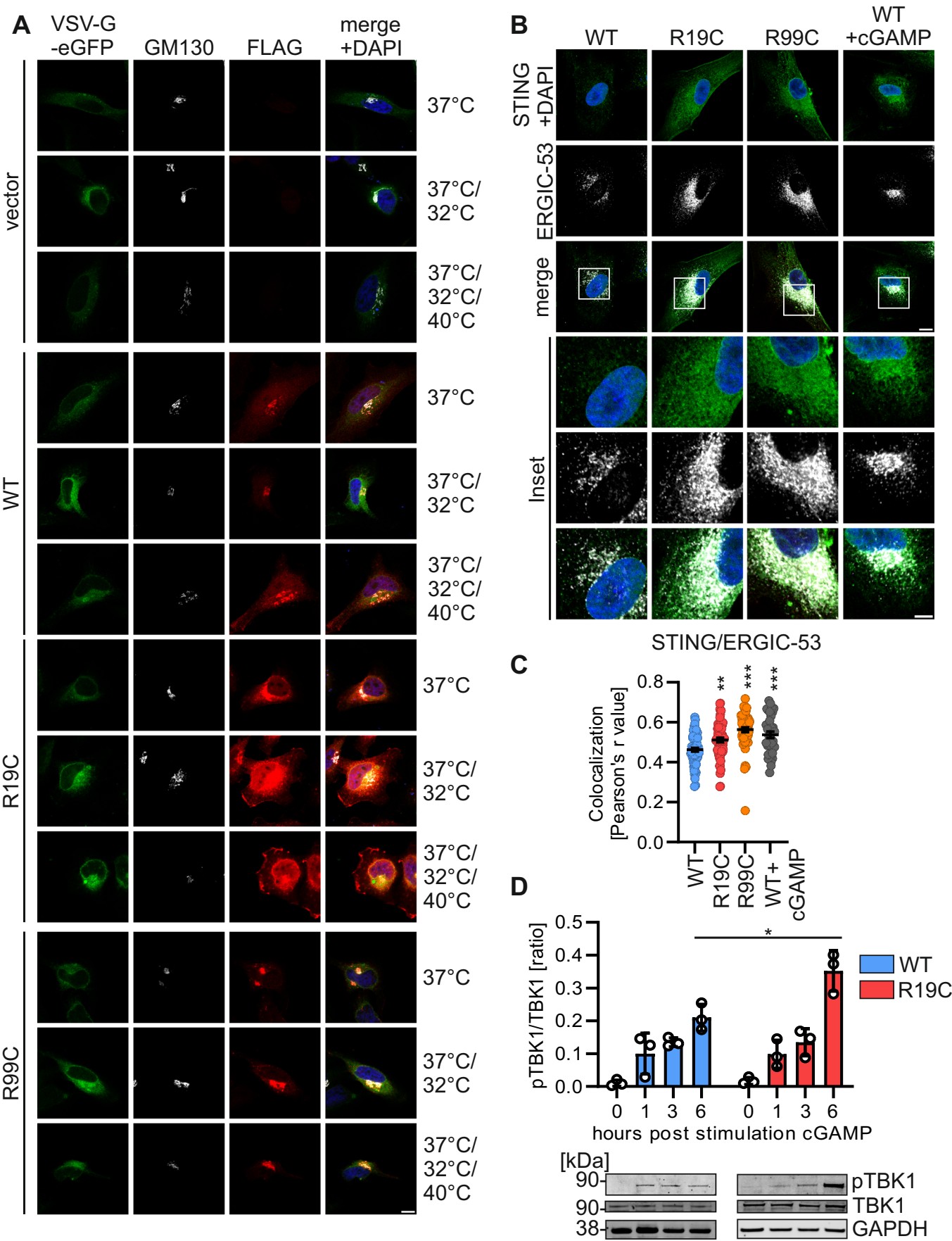

◄ **Figure EV3. ARF1 R19C impairs retrograde transport and accumulates STING at the ERGIC.**

(A) Representative immunofluorescence images of Hela cells expressing ARF1 WT, R19C, R99C or vector control together with VSV-G-eGFP (green). 24 h post transfection the cells were subjected to the indicated temperature shifts (37 °C/32 °C/40 °C) and stained with anti-FLAG (red) and anti-GM130 (white). Nuclei: DAPI (blue). Scale bar: 10 μm. (B) Representative immunofluorescence images of primary fibroblasts of a healthy donor (WT), patient AGS3238 (R19C) or patient AGS460 (R99C). Healthy donor fibroblasts treated with cGAMP (20 μg/ml, 2 h) were used as positive control. Cells were stained with anti-STING (green) and anti-ERGIC-53 (white). Nuclei: DAPI (blue). Insets are shown in higher magnification. Scale bar: 10 μm (full size images), 5 μm (insets). (C) Colocalization (Pearson's correlation coefficient, r) between STING and ERGIC-53 from the images shown in (B). Lines represent the mean of $n = 33$–77 ± SEM (individual cells). Statistical analysis was performed using two-tailed Student's t test with Welch's correction. **$p < 0.01$ ($p = 0.0010$ WT vs R19C); ***$p < 0.001$ ($p < 0.0001$ WT vs R99C, WT vs cGAMP). (D) Quantification of the pTBK1 band intensities normalized to TBK1 band intensities in primary fibroblasts from a healthy donor (WT) or patient AGS3238 (R19C) at the indicated time points post stimulation with cGAMP (20 μg/ml). Bars represent mean of $n = 3$ ± SEM (biological replicates). Statistical analysis was performed using two-tailed Student's t test with Welch's correction. *$p < 0.05$ ($p = 0.0375$ WT vs R19C (6 h)). Blots were stained with anti-pTBK1, anti-TBK1 and anti-GAPDH. Source data are available online for this figure.

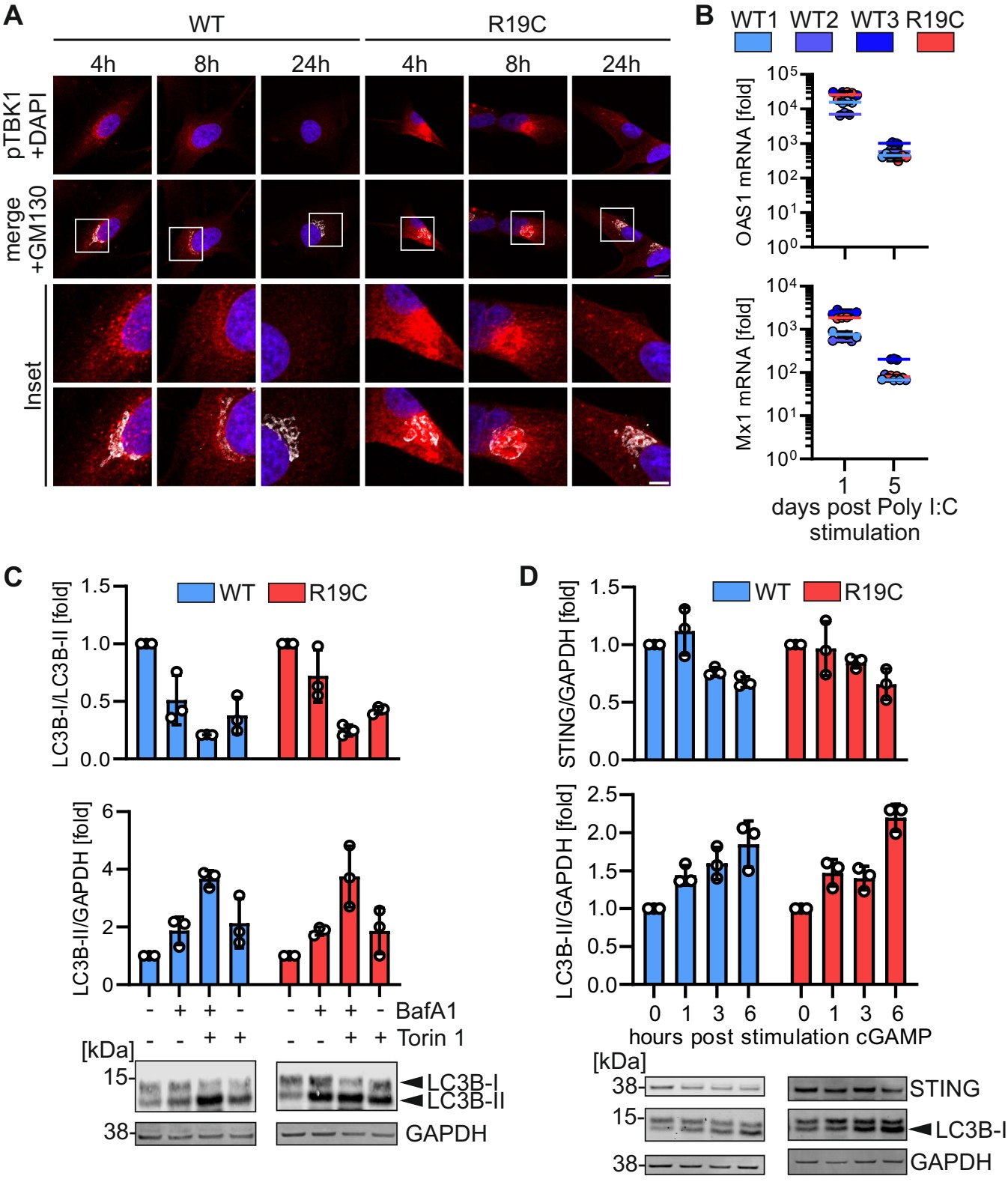

**Figure EV4.** **Impact of cGAMP-stimulation on TBK1 activation and STING degradation.**

(A) Representative immunofluorescence images of primary fibroblasts of a healthy donor (WT) or patient AGS3238 (R19C). Cells were treated with cGAMP (50 µg/ml) and stained at the indicated timepoints post stimulation with anti-pTBK1 (red) and anti-GM130 (white). Nuclei: DAPI (blue). Insets are shown in higher magnification. Scale bar: 10 µm (full size images), 5 µm (insets). (B) fold induction of OAS1 (top) and Mx1 (bottom) mRNA levels in primary fibroblasts from healthy donors (WT) or patient AGS3238 (R19C) as assessed by qPCR at indicated time points following stimulation with Poly I:C (1 µg/ml). Lines represent the mean of $n = 3 \pm$ SEM (biological replicates). (C) Quantification of the LC3B-I/LC3B-II (top) or LC3B-II/GAPDH (bottom) band intensities in primary fibroblasts from a healthy donor (WT) or patient AGS3238 (R19C) 4 h post treatment with bafilomycin A1 (BafA1, 250 nM), Torin 1 (1 µM) or both. Bars represent mean of $n = 3 \pm$ SEM (biological replicates). Blots were stained with anti-LC3B and anti-GAPDH. (D) Quantification of the STING (top) or LC3B-II (bottom) band intensities normalized to GAPDH band intensities in primary fibroblasts from a healthy donor (WT) or patient AGS3238 (R19C) at the indicated time points post stimulation with cGAMP (20 µg/ml). Bars represent mean of $n = 3 \pm$ SEM (biological replicates). Blots were stained with anti-STING, anti-LC3B and anti-GAPDH.

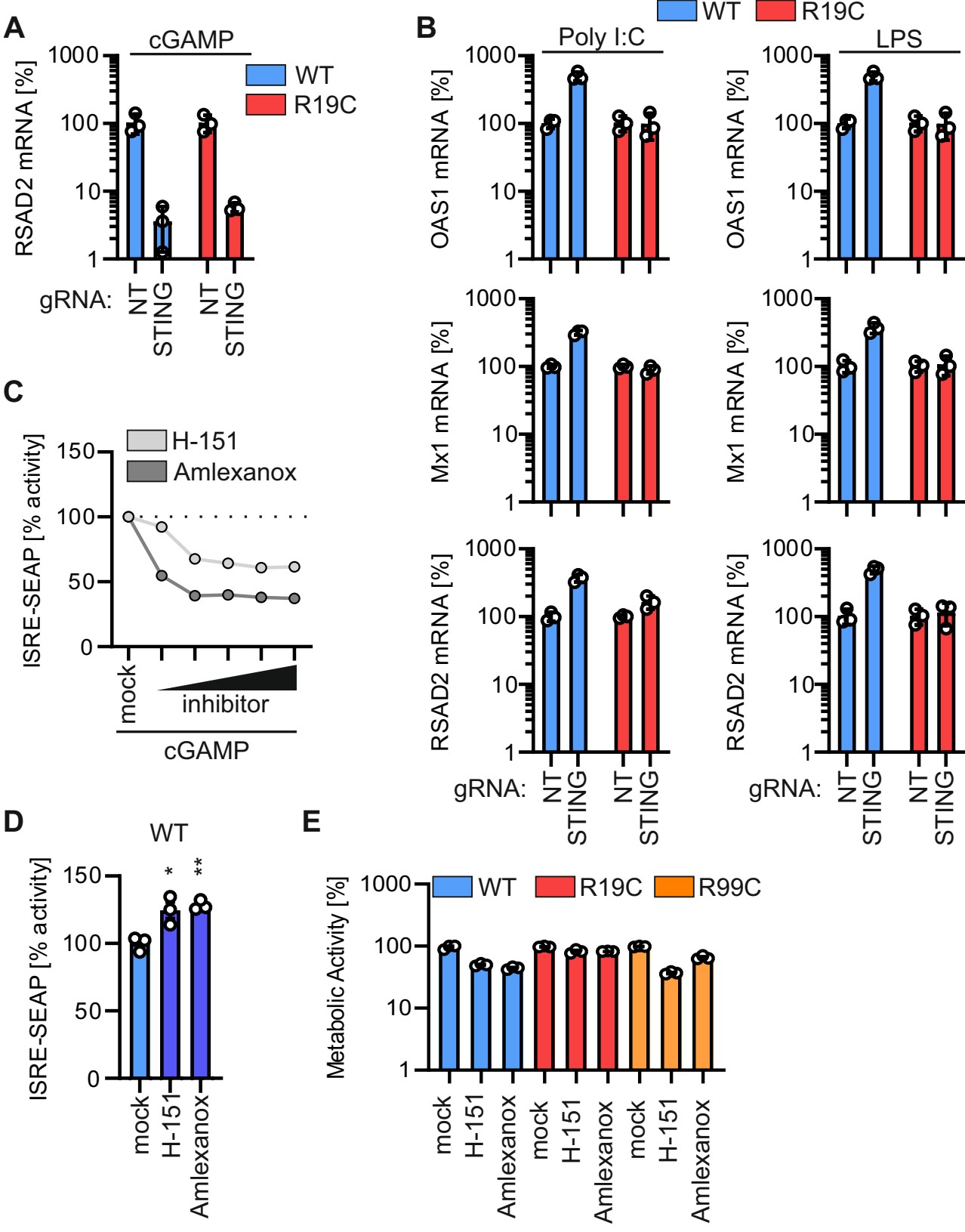

◄  **Figure EV5.   Dysregulated type I IFN signalling is specific for the cGAS-STING pathway and treatment approaches.**

(A) fold induction of RSAD2 mRNA levels in primary fibroblasts from a healthy donor (WT) or patient AGS3238 (R19C) as assessed by qPCR 16 h post stimulation with cGAMP (20 µg/ml). Treatment was performed 96 h post electroporation with Cas9/gRNA RNPs targeting either STING or non-targeting (NT). Lines represent the mean of $n = 3 \pm$ SEM (biological replicates). (B) fold induction of Oas1 (top), Mx1 (middle) and RSAD2 (bottom) mRNA levels in primary fibroblasts from a healthy donor (WT) or patient AGS3238 (R19C) as assessed by qPCR 16 h post stimulation with Poly I:C (1 µg/ml, left panels) or LPS (5 µg/ml, right panels). Treatment was performed 96 h post electroporation with Cas9-gRNA complexes targeting either STING or non-targeting (NT). Lines represent the mean of $n = 3 \pm$ SEM (biological replicates). (C) ISRE-SEAP activity of HEK293-STING cells stimulated with 20 µg/ml cGAMP (2 h prior to H 151, 1 h prior to amlexanox) and treated with increasing concentrations of H-151 (0.1 µM, 1 µM, 2 µM, 5 µM, 10 µM) or amlexanox (10 µg/ml, 20 µg/ml, 33 µg/ml, 40 µg/ml, 50 µg/ml). SEAP activity was quantified 16 h post treatment and normalized to cGAMP-stimulated control (mock). Dots present the mean of $n = 3 \pm$ SEM (biological replicates). (D) Supernatant transfer from primary fibroblasts of a healthy donor (WT) consecutively treated four times every 48 h with H-151 (2 µM) or amlexanox (33 µg/ml) to HEK293-STING cells (293-Dual-hSTING-R232). ISRE-SEAP activity was quantified 48 h post transfer and normalised to metabolic activity. Lines represent the mean of $n = 3 \pm$ SEM (biological replicates). Statistical analysis was performed using two-tailed Student's t test with Welch's correction. *$p < 0.05$ ($p = 0.0361$ mock vs H-151); **$p < 0.01$ ($p = 0.0028$ mock vs amlexanox). (E) Cell viability of the primary fibroblasts from Figs. 5G, H and EV3D prior to supernatant transfer as assessed by MTT assay and normalized to the respective mock control. Lines represent the mean of $n = 3 \pm$ SEM (biological replicates).

