## [Peer Review File · EMBO Reports]

Distinct pathogenic mutations in ARF1 allow dissection of its dual role in cGAS-STING signaling

Johannes Lang, Tim Bergner, Julia Zinngrebe, Alice Lepelley, Katharina Vill, Steffen Leiz, Meinhard Wlaschek, Matias Wagner, Karin Scharffetter-Kochanek, Pamela Fischer-Posovszky, Clarissa Read, Yanick Crow, Maximilian Hirschenberger, and Konstantin Sparrer

Corresponding author(s): Konstantin Sparrer (konstantin.sparrer@uni-ulm.de) , Maximilian Hirschenberger (maximilian.hirschenberger@uni-ulm.de)

Review Timeline:

Submission Date:	15th Aug 24
Editorial Decision:	20th Sep 24
Revision Received:	19th Dec 24
Editorial Decision:	21st Jan 25
Revision Received:	19th Feb 25
Accepted:	27th Feb 25

Transaction Report:

Dear Dr. Sparrer

Thank you for the submission of your research manuscript to our journal. We have now received the full set of referee reports that is copied below.

As you will see, all three referees acknowledge that the findings are interesting, but they also raise a number of - partially overlapping - concerns and have suggestions how to further strengthen the data. I think that all of the concerns are relevant and should be addressed. Please let me know in case you disagree, and we can discuss the exact revision requirements further, also in a video chat, if you like.

Given these constructive comments, we would like to invite you to revise your manuscript with the understanding that the referee concerns (as detailed above and in their reports) must be fully addressed and their suggestions taken on board. Please address all referee concerns in a complete point-by-point response. Acceptance of the manuscript will depend on a positive outcome of a second round of review. It is EMBO Reports policy to allow a single round of revision only and acceptance or rejection of the manuscript will therefore depend on the completeness of your responses included in the next, final version of the manuscript.

We realize that it is difficult to revise to a specific deadline. In the interest of protecting the conceptual advance provided by the work, we recommend a revision within 3 months (December 20). Please discuss the revision progress ahead of this time with the editor if you require more time to complete the revisions.

*****IMPORTANT NOTE:

We perform an initial quality control of all revised manuscripts before re-review. Your manuscript will FAIL this control and the handling will be delayed IN CASE the following APPLIES:

- 1) A data availability section providing access to data deposited in public databases is missing. If you have not deposited any data, please add a sentence to the data availability section that explains that.
- 2) Your manuscript contains statistics and error bars based on $n=2$. Please use scatter blots in these cases. No statistics should be calculated if $n=2$.

When submitting your revised manuscript, please carefully review the instructions that follow below. Failure to include requested items will delay the evaluation of your revision.*****

- 1) a .docx formatted version of the manuscript text (including legends for main figures, EV figures and tables). Please make sure that the changes are highlighted to be clearly visible.
- 2) individual production quality figure files as .eps, .tif, .jpg (one file per figure). Please download our Figure Preparation Guidelines (figure preparation pdf) from our Author Guidelines pages <https://www.embopress.org/page/journal/14693178/authorguide> for more info on how to prepare your figures.
- 3) a .docx formatted letter INCLUDING the reviewers' reports and your detailed point-by-point responses to their comments. As part of the EMBO Press transparent editorial process, the point-by-point response is part of the Review Process File (RPF), which will be published alongside your paper.
- 4) a complete author checklist, which you can download from our author guidelines (<<https://www.embopress.org/page/journal/14693178/authorguide>>). Please insert information in the checklist that is also reflected in the manuscript. The completed author checklist will also be part of the RPF.
- 5) Please note that all corresponding authors are required to supply an ORCID ID for their name upon submission of a revised manuscript (<<https://orcid.org/>>). Please find instructions on how to link your ORCID ID to your account in our manuscript

tracking system in our Author guidelines

(<<https://www.embopress.org/page/journal/14693178/authorguide#authorshipguidelines>>)

6) We replaced Supplementary Information with Expanded View (EV) Figures and Tables that are collapsible/expandable online. A maximum of 5 EV Figures can be typeset. EV Figures should be cited as 'Figure EV1, Figure EV2' etc... in the text and their respective legends should be included in the main text after the legends of regular figures.

7) Please include a dedicated "Data Availability" section at the end of the Methods (suggested wording: "The [structural coordinates | microarray | mass spectrometry] data from this publication have been deposited to the [name of the database] database [URL] and assigned the identifier [accession | permalink | hashtag]."). Should this not apply, this should still be stated as "This study includes no data deposited in external repositories."

Additional information on source data and instruction on how to label the files are available

<<https://www.embopress.org/page/journal/14693178/authorguide#sourcedata>>.

10) Figure legends and data quantification:

- the name of the statistical test used to generate error bars and P values,
- the number (n) of independent experiments (please specify technical or biological replicates) underlying each data point,
- the nature of the bars and error bars (s.d., s.e.m.)

- If the data are obtained from n {less than or equal to} 5, show the individual data points in addition to the SD or SEM.

- If the data are obtained from n {less than or equal to} 2, use scatter blots showing the individual data points.

11) Our journal encourages inclusion of *data citations in the reference list* to directly cite datasets that were re-used and obtained from public databases. Data citations in the article text are distinct from normal bibliographical citations and should directly link to the database records from which the data can be accessed. In the main text, data citations are formatted as follows: "Data ref: Smith et al, 2001" or "Data ref: NCBI Sequence Read Archive PRJNA342805, 2017". In the Reference list, data citations must be labeled with "[DATASET]". A data reference must provide the database name, accession number/identifiers and a resolvable link to the landing page from which the data can be accessed at the end of the reference. Further instructions are available at <<https://www.embopress.org/page/journal/14693178/authorguide#referencesformat>>.

12) All Materials and Methods need to be described in the main text using our 'Structured Methods' format. According to this format, the Methods section includes a Reagents and Tools Table (listing key reagents, experimental models, software and relevant equipment and including their sources and relevant identifiers) followed by a Methods and Protocols section describing the methods, ideally using a step-by-step protocol format. The aim is to facilitate adoption of the methodologies across labs. Please download and fill our Reagents and Tools Table template (.docx), which you can find in our author guidelines: <https://www.embopress.org/page/journal/14693178/authorguide#structuredmethods>.

13) As part of the EMBO publication's Transparent Editorial Process, EMBO Reports publishes online a Review Process File to accompany accepted manuscripts. This File will be published in conjunction with your paper and will include the referee reports, your point-by-point response and all pertinent correspondence relating to the manuscript.

Yours sincerely,

=====

Referee #1:

In this manuscript, the authors provided the clear evidence that the ARF1 mutation-dependent interferon upregulation is solely caused by the impaired retrograde membrane trafficking from the Golgi to the ER by exploiting a new ARF1 variant (R19C) found in a patient with type I interferonopathy. The study is well designed and the results are overall convincing. This reviewer has a few suggestions to improve the quality of the study.

Major critique:

(1) The authors demonstrated that STING activation in cells expressing R19C did not depend on mitochondrial DNA release, which is neat. I ask the authors to investigate the role of cGAS in these cells. A few papers suggested the dispensable role of cGAS in STING activation in cells with COPA mutations (PMID: 33397928, 38212328), which also interferes with the retrograde trafficking. Thus, this issue is interesting and needs to be addressed. The paragraph in Discussion (from lines 283) can be revised, according to the results.

Minor critiques:

(2) line 41: "cytoplasmic" should read "cytosolic".

(3) The characterization of AGS3238, such as the data of DNA sequencing and MRI scan, should be provided.

(4) line 234: H-151 can be described as "the STING palmitoylation inhibitor".

Referee #2:

Lang et al report a novel disease-causing variant, R19C in ARF1. The authors propose mutation of R19 to C decreases ARF1 GTPase activity and disrupts ARF1 interaction with the COP-I complex, leading to STING driven inflammation. This is an interesting finding, important for the autoinflammatory disease community and is worthy of publication. This study is highly appropriate for EMBO reports.

Analysis of protein trafficking defects and GTPase activity are convincing. However, the interrogation of spontaneous inflammatory signalling is less robust and requires additional experiments and/or a more refined interpretation. It is not clear from the data presented if STING spontaneously signals in R19C ARF1 cells or STING signalling is enhanced and prolonged but still needs activation by cGas or another inflammatory pathway (eg: any induction of IFN α increasing levels of STING). It is possible that the R19C ARF1 variant enhances a polygenetic phenotype in this patient. This manuscript would be markedly improved by

the addition of a few simple experiments with more direct read outs. At a minimum, the following experiments should be done:

1. Isogenic correction of the R19C ARF1 variant in patient cells (eg: low passage number fibroblasts) and analysis of ISG transcription. This is particularly important because later figures have only used a single healthy donor for comparison (eg: Figure 4C).
2. Gene deletion of cGas and STING in patient cells. As per Figure 1A, there is a robust ISG signature in patient cells (I note that it is unclear from the legend if these are PBMCs or fibroblasts). To determine if this is only dependant on STING, CRISPR/Cas9 delete cGas and STING in patient cells (and control lines) and directly measure a panel of ISGs. Include controls external to the cGas/STING pathway, e.g.: interferon inducing pattern recognition receptors (PRRs) such as TLR7, RIG-I or PKR. This will allow the authors to infer the relative importance of STING, cGas and other PRRs to R19C ARF1 associated inflammation.

Please note, in both cases I ask for a direct read out such as ISG transcription in patient cells, this is in contrast to the supernatant transfer to the reporter cells used eg: Figure 5F-G, which is an indirect readout and can be confounded by events such as inhibitor toxicity on patient (or other) cells. Viability data presented in Fig EV3 is difficult to interpret and shows no error bars, it does not allow comparison of viability between lines (continual STING activation can trigger cell death). Showing cell viability by Propidium Iodine staining by flow cytometry and including representative scatter plots would significantly improve this.

Major Comments:

3. Figure 2B, "In line with this, mtDNA levels in the cytoplasm upon expression of ARF1 R19C in HEK293T were comparable or even lower than the WT or vector control (Fig. 2B)." This is not reflected in the Tfam western blot. R19C looks equivalent to the WT and R99C looks less. Accepting that this is a technically challenging experiment where low levels of proteins would be sensitive to experimental variance, I do not think the author's conclusion is necessarily incorrect (ie: R19C ARF1 does not disrupt mitochondrial health) but it needs to be confirmed with better evidence. If a clearer western blot cannot be produced the authors could:
 - a. -Include other mitochondrial genes in Figure 2C to consistently show limited mtDNA in the cytosol in R19C compared to R99C (also, in figure legend please state whether the 6 replicates are from independent experiments).
 - b. -Immunoprecipitate cGas from these samples and probe for mitochondrial genes directly (as per White et al 2014). This would also address other review points.
4. Figure 4A: it's interesting to note that R19C ARF1-FLAG staining is diffuse throughout the cell compared to WT and R99C which localise more closely with GM130 staining. The diffuse distribution of R19C is similar to what was observed in the ARF1 T31N mutant which is trapped in the GDP bound state, as previously used in a different study by this group (Hirschenberger et al). Unlike R19C, T31N did not trigger inflammation or disrupt STING localisation, even in the context of STING over expression. A discussion point comparing the two mutants (R19C and T31N) should be added.
5. Figure 5A-C: Does the level of STING persist over time in patient cells? STING signalling is terminated via lysosomal degradation (e.g.: Gentili et al 2023) and ARF1 may play a role in autophagy regulation (eg: van der Vaart et al 2010). Could there be a defect in STING degradation in R19C ARF1 cells which explains the prolonged signalling? Quantification of STING levels in immunofluorescence samples already obtained should be shown. To extend this, the authors could do simple western blots comparing LC3-I and LC3-II levels or immunofluorescence for colocalization of CD63, p62 and/or LC3B and STING (eg: as per Gentili et al 2023).

Minor Comments:

6. A figure showing patient phenotype should be included.
7. Figure 1D, there is no WT curve shown
8. Figure 1C and D: Increasing concentrations of STING can spontaneously trigger signalling, and this may be negatively regulated by WT ARF1, therefore an empty vector control for ARF1 should be included in these panels.
9. Figure 2A, please provide quantification of mitochondrial stress (or lack thereof) or assess mitochondrial health by measuring mitochondrial membrane potential (TMRE staining), mitochondria ROS (mitoSOX) or mtDNA release.
10. Figure 2D: The lack of induction of IFN in these cells could be due to lower STING levels (as suggested), or it could be due to these cells being cGas negative (as previously reported for models of COPA syndrome (Lepelley et al, 2020 and Steiner et al, 2022)). Depending on the results of above suggested experiments the authors should highlight this in the text.
11. Figure 2E and F: please show untreated controls for context. It is not clear from the legend if each sample is normalised to its untreated control or both are normalised to the untreated healthy donor. In this case, I suggest normalising fold change to untreated healthy donor would be most appropriate to demonstrate 1) basal elevation of ISGs such as Mx1 and OAS1 in patient fibroblasts and 2) the stronger response to STING stimulation. Presenting the data in this way would help in interpretation of the overall effect of the R19C ARF1 variant.

12. Additionally, the results of Figure 2E and F should not be over interpreted, basal IFN signalling elevates proteins essential for IFN signalling (eg: STAT1) and therefore these cells may be hyper responsive to all IFN inducing stimuli. Inclusion of other PRR agonists would determine whether the elevated ISG induction is specific to STING signalling or a generalised effect.

13. Figure 3A: Golgi disorganisation should be quantified

14. Figure 3B: why was COPg the only COP-I subunit blotted for? A sentence in text would improve manuscript readability.

15. Figure 4E and 5C: these data should be backed up by a phosphorylated TBK1 western blots.

16. Where possible addition of more healthy donors for comparison would improve the robustness of this study (eg: Figure 2E-F; Figure 4C, etc) as basal ISG levels (etc) will vary between individuals. I commend the comparison of 29 healthy donors in Figure 1A and hope the authors can extend this to include additional donors in fibroblast experiments.

Text:

Overall the text is fine, there are a few sentences that don't read well for me (eg: line 50: disrupt should be disrupts, line 122: starting a new paragraph with a joining phrase "in line" is confusing) but this should all be picked up in the editing process. I have picked up a couple phrases which need to be rewritten for scientific clarity:

Line 147: "In line, in 293-Dual hSTING-R232 cells, which endogenously express human STING" should be rewritten, these do not endogenously express STING. Exogenous STING has been stably expressed in this line.

Line 199: "To terminate DNA sensing, STING is transported in an ARF1-dependent manner from the Golgi to the ER." I understand the gist of this sentence, but it should be rewritten, STING does not directly sense DNA and STING signalling can be halted by transport to lysosomes for degradation.

More details in figure legends would improve readability of this manuscript.

Overall, I commend the authors on their work and wish them luck with their follow up experiments.

Referee #3:

This manuscript by Lang et al is a follow up to their work published last year in Nature Communications (PMID 37914730), which described a disease-causing mutation (R99) in the GTPase ARF1, thereby driving type I IFN-driven pathology. This current manuscript follows the same template in characterising a separate mutation (R19C), performing many of the same assays which were used in the previous paper. While the novelty is limited given the similarities to their previous work, this is still an elegant, well-conducted piece of work which has also shed further light on the dual role of ARF1 in regulating mitochondrial dynamics and termination of STING signalling. I have some minor suggestions as to how the manuscript might be improved, which are listed below:

1. In line 213, the authors claim to have provided evidence that 'active' STING accumulates at the ERGIC/GOLGI. However, active STING would indicate that they have measured levels of phosphorylated STING, which they have not. The authors could measure p-STING in the immunofluorescence experiments they have performed or simply measure it by Western Blot in their mutant fibroblasts.

2. The immunoprecipitation experiments (Fig. 3B) are important in demonstrating a reduced ARF1-COPI interaction in mutant cells. However, the authors performed this experiment in an overexpression system which might be considered slightly artificial. Given the importance of this experiment, would it be possible to perform the immunoprecipitation endogenously in patient fibroblasts?

3. Fig. 2D is the only figure which I have a slight conceptual issue with. The data derived from patient samples (Fig. 1A and Figs 5F and G) show the presence of an interferon response in R19C even in the absence of stimulation, yet experiments in the hSTING-R232 cells indicate that STING must be active in order to obtain an interferon response in the R19C mutant. I assume that the difference owes to the primary human cells already exhibiting a basal level of active STING. However, some readers might find these results slightly contradictory, so it may be worth including a line of discussion concerning this point.

4. In lines 129-132, the authors compare the doses of the respective mutant plasmids which are required to induce IFN signalling. I think that it is tricky to make this claim given that increased plasmid dose does not necessarily equal increased expression as expression patterns may differ between plasmids. I would suggest the authors remove this line.

5. It would be great if the authors could include electron microscopy images of the R99C mutant alongside the WT and R19C mutant in Figure 2A for comparison

6. Along the same lines, could the authors move the R99C colocalization data (Fig. EV3B) alongside the R19C colocalization data in Fig. 3D for comparison?

7. Fig. 5D shows an increased IFN response in the mutant patient fibroblasts when stimulated with cGAMP. As an additional control to demonstrate that cGAS-STING signalling specifically is heightened in these cells, could the authors also stimulate the cells with a different PRR agonist (e.g. LPS)? Theoretically, LPS-induced responses should be similar between wild-type and mutants

**Point-by-point response to reviewer comments to manuscript no. EMBOR-2024-60207V1
“Distinct pathogenic mutations in ARF1 allow dissection of its dual role in cGAS-STING signaling”**

Referee #1:

In this manuscript, the authors provided the clear evidence that the ARF1 mutation-dependent interferon upregulation is solely caused by the impaired retrograde membrane trafficking from the Golgi to the ER by exploiting a new ARF1 variant (R19C) found in a patient with type I interferonopathy. The study is well designed and the results are overall convincing. This reviewer has a few suggestions to improve the quality of the study.

We thank the reviewer for his/her positive evaluation of our manuscript.

Major critique:

(1) The authors demonstrated that STING activation in cells expressing R19C did not depend on mitochondrial DNA release, which is neat. I ask the authors to investigate the role of cGAS in these cells. A few papers suggested the dispensable role of cGAS in STING activation in cells with COPA mutations (PMID: 33397928, 38212328), which also interferes with the retrograde trafficking. Thus, this issue is interesting and needs to be addressed. The paragraph in Discussion (from lines 283) can be revised, according to the results.

This is indeed a relevant question. Our data shows that, despite the presence of endogenously expressed STING, overexpression of ARF1 R19C does not activate type I IFN signalling (Fig. 2D). This suggests that inflammation is not induced in the absence of a trigger by ARF1 R19C, as opposed to ARF1 R99C which provides its ‘own’ cGAS ligands by destabilizing mitochondria. We have amended our discussion to make this point clearer and cite the relevant literature (lines 272-274, lines 300-301 and line 305-315).

Minor critiques:

(2) line 41: "cytoplasmic" should read "cytosolic".

Done.

(3) The characterization of AGS3238, such as the data of DNA sequencing and MRI scan, should be provided.

The Informed Consent unfortunately does not permit the publication of these datasets. We have updated the patient description with more details, as far as the local ethics committee has permitted us to do so.

(4) line 234: H-151 can be described as "the STING palmitoylation inhibitor".

This has been added.

Referee #2:

Lang et al report a novel disease-causing variant, R19C in ARF1. The authors propose mutation of R19 to C decreases ARF1 GTPase activity and disrupts ARF1 interaction with the COP-I complex, leading to STING driven inflammation. This is an interesting finding, important for the autoinflammatory disease community and is worthy of publication. This study is highly appropriate for EMBO reports.

We thank this reviewer for the highly positive comment.

Analysis of protein trafficking defects and GTPase activity are convincing. However, the interrogation of spontaneous inflammatory signalling is less robust and requires additional experiments and/or a more refined interpretation. It is not clear from the data presented if STING spontaneously signals in R19C ARF1 cells or STING signalling is enhanced and prolonged but still needs activation by cGas or another inflammatory pathway (eg: any induction of IFN α increasing levels of STING). It is possible that the R19C ARF1 variant enhances a polygenetic phenotype in this patient. This manuscript would be markedly improved by the addition of a few simple experiments with more direct read outs. At a minimum, the following experiments should be done:

1. Isogenic correction of the R19C ARF1 variant in patient cells (eg: low passage number fibroblasts) and analysis of ISG transcription. This is particularly important because later figures have only used a single healthy donor for comparison (eg: Figure 4C).

Our assays showed that sustained signalling in the presence of ARF1 R19C requires stimulation of the cGAS/STING pathway, either by STING overexpression or cGAMP treatment (see Fig. 1C, updated Fig. 2E,F). In the presence of endogenous STING, expression of ARF19C alone does not activate type I IFN signalling (Fig. 2D). We have amended our discussion to include this important point (line 300-301).

We have now included 2 more low passage fibroblast donors, repeating all experiments with three donors comparing to the patient fibroblasts and analysing ISG transcription (new Fig. 2 E, F; new Fig. 5 D,E; Fig. EV4 B, C).

2. Gene deletion of cGas and STING in patient cells. As per Figure 1A, there is a robust ISG signature in patient cells (I note that it is unclear from the legend if these are PBMCs or fibroblasts). To determine if this is only dependant on STING, CRISPR/Cas9 delete cGas and STING in patient cells (and control lines) and directly measure a panel of ISGs. Include controls external to the cGas/STING pathway, e.g.: interferon inducing pattern recognition receptors (PRRs) such as TLR7, RIG-I or PKR. This will allow the authors to infer the relative importance of STING, cGas and other PRRs to R19C ARF1 associated inflammation. Please note, in both cases I ask for a direct read out such as ISG transcription in patient cells, this is in contrast to the supernatant transfer to the reporter cells used eg: Figure 5F-G, which is an indirect readout and can be confounded by events such as inhibitor toxicity on patient (or other) cells.

PBMC data has been added to the Figure 1A. To address the reviewer's concern, we have now established a technically challenging new protocol to perform Cas9/gRNA RNP KOs using electroporation directly in primary fibroblasts in one step. Of note, the primary cells are only passaged <5 times, thus KO lines cannot be produced. Our data shows that depletion of STING reduces endogenous ISG levels after cGAMP stimulation, but has little impact on ISG transcription following polyI:C (RIG-I) or LPS (TLR4) stimulation (new Fig. EV5B; new Fig. 5F). Gene depletion of cGAS using CRISPR/Cas9 was unfortunately unsuccessful.

Viability data presented in Fig EV3 is difficult to interpret and shows no error bars, it does not allow comparison of viability between lines (continual STING activation can trigger cell death). Showing cell viability by Propidium Iodine staining by flow cytometry and including representative scatter plots would significantly improve this.

We now show the data as bar graphs including error bars (updated Fig. EV5E).

Major Comments:

3. Figure 2B, "In line with this, mtDNA levels in the cytoplasm upon expression of ARF1 R19C in HEK293T were comparable or even lower than the WT or vector control (Fig. 2B.." This is not reflected in the Tfam western blot. R19C looks equivalent to the WT and R99C looks less. Accepting

that this is a technically challenging experiment where low levels of proteins would be sensitive to experimental variance, I do not think the author's conclusion is necessarily incorrect (ie: R19C ARF1 does not disrupt mitochondrial health) but it needs to be confirmed with better evidence. If a clearer western blot cannot be produced the authors could:

- a. -Include other mitochondrial genes in Figure 2C to consistently show limited mtDNA in the cytosol in R19C compared to R99C (also, in figure legend please state whether the 6 replicates are from independent experiments).
- b. -Immunoprecipitate cGas from these samples and probe for mitochondrial genes directly (as per White et al 2014). This would also address other review points.

This experiment is indeed technically very challenging. However, we show that other mitochondrial genes (as assessed by qPCR, MT-ND1) behave similarly (new Fig. EV1 C). All experiments are the combination of 6 independent, biological replicates.

4. Figure 4A: it's interesting to note that R19C ARF1-FLAG staining is diffuse throughout the cell compared to WT and R99C which localise more closely with GM130 staining. The diffuse distribution of R19C is similar to what was observed in the ARF1 T31N mutant which is trapped in the GDP bound state, as previously used in a different study by this group (Hirschenberger et al). Unlike R19C, T31N did not trigger inflammation or disrupt STING localisation, even in the context of STING over expression. A discussion point comparing the two mutants (R19C and T31N) should be added.

This is an interesting point, however, our quantitative analysis of the co-localisation with GM130 shows that ARF1 R19C localises more to the Golgi than WT ARF1, but less than R99C ARF1 (Fig. 4A and Fig. EV2 A,B), thus not being similar to ARF1 T31N.

5. Figure 5A-C: Does the level of STING persist over time in patient cells? STING signalling is terminated via lysosomal degradation (e.g.: Gentili et al 2023) and ARF1 may play a role in autophagy regulation (eg: van der Vaart et al 2010). Could there be a defect in STING degradation in R19C ARF1 cells which explains the prolonged signalling? Quantification of STING levels in immunofluorescence samples already obtained should be shown. To extend this, the authors could do simple western blots comparing LC3-I and LC3-II levels or immunofluorescence for colocalization of CD63, p62 and/or LC3B and STING (eg: as per Gentili et al 2023).

As the reviewer suggests, we have performed western blot analyses of primary cells staining endogenous STING and LC3B. Upon stimulation with cGAMP our data shows that both degradation of STING and induction of autophagy (as visualized by LC3B processing) are similar in WT and patient cells in three independent experiments (new Fig. EV4D). This suggests that there is no defect in degradation. We opted to not quantify STING levels (e.g. by mean fluorescence) in the immunofluorescent images, to avoid the bias of comparing cells with larger/smaller cytoplasm, and concerns that the signal/noise ratio of the antibody used was insufficient for robust quantification.

Minor Comments:

6. A figure showing patient phenotype should be included.

The Informed Consent does not permit the publication of these datasets. We have, however, amended the patient information with as much detail as we are allowed to provide.

7. Figure 1D, there is no WT curve shown

The curve for ARF1 WT has been added (updated Fig. 1D).

8. Figure 1C and D: Increasing concentrations of STING can spontaneously trigger signalling, and

this may be negatively regulated by WT ARF1, therefore an empty vector control for ARF1 should be included in these panels.

Empty vector controls (e.g. STING + EV) were included (see updated Fig. EV1 A, B, E)

9. Figure 2A, please provide quantification of mitochondrial stress (or lack thereof) or assess mitochondrial health by measuring mitochondrial membrane potential (TMRE staining), mitochondria ROS (mitoSOX) or mtDNA release.

We have now quantified the mitochondrial membrane potential measured as MFI of mitotracker Red / MFI of mitotracker Green (new figure EV1 D). In line with the qPCR analysis and the EM, this data shows that the mitochondria are intact in the presence of ARF1 R19C.

10. Figure 2D: The lack of induction of IFN in these cells could be due to lower STING levels (as suggested), or it could be due to these cells being cGas negative (as previously reported for models of COPA syndrome (Lepelley et al, 2020 and Steiner et al, 2022)). Depending on the results of above suggested experiments the authors should highlight this in the text.

The cells used in the experiment are constitutively STING expressing HEK293T variants (commercially available HEK293T cells). Indeed, cGAS levels are expected to be low in HEK293T. We have clarified these points in the results section (lines 138-50).

11. Figure 2E and F: please show untreated controls for context. It is not clear from the legend if each sample is normalised to its untreated control or both are normalised to the untreated healthy donor. In this case, I suggest normalising fold change to untreated healthy donor would be most appropriate to demonstrate 1) basal elevation of ISGs such as Mx1 and OAS1 in patient fibroblasts and 2) the stronger response to STING stimulation. Presenting the data in this way would help in interpretation of the overall effect of the R19C ARF1 variant.

Since we are now using multiple individual donors for these experiments, we normalized each donor to the corresponding untreated condition (new Fig. 2E, F) showing stronger response to STING stimulation. The basal ISG levels of the patient fibroblasts do not show an IFN signature compared to healthy donors, unlike the PBMCs (Fig. 1A). We assume this is either due to the fibroblasts not being challenged/activated before, unlike PBMCs, or that PBMCs in general react more sensitively to type I IFN. We have discussed this point now in our discussion (line 307-311).

12. Additionally, the results of Figure 2E and F should not be over interpreted, basal IFN signalling elevates proteins essential for IFN signalling (eg: STAT1) and therefore these cells may be hyper responsive to all IFN inducing stimuli. Inclusion of other PRR agonists would determine whether the elevated ISG induction is specific to STING signalling or a generalised effect.

Our new data shows that ARF1 R19C patient cells show comparable response to PolyI:C as three healthy donors (new Fig. EV4 B, C). Elevation of ISG levels in patient fibroblasts is specific to cGAMP treatment (new Fig. 5F and new Fig. EV4 B, C). Of note, STING KO does not alter induction of ISGs following polyI:C or LPS stimulation in patient cells (new Fig. EV5B).

13. Figure 3A: Golgi disorganisation should be quantified

This has been added as new Fig. 3B.

14. Figure 3B: why was COPg the only COP-I subunit blotted for? A sentence in text would improve manuscript readability.

COPg is a defining protein of COPI complexes, and previous structural data has shown that ARF1 binds to COPg (PDB: 3TJZ and Yu et al, 2012), we have clarified this in the results section now (lines 167-169).

15. Figure 4E and 5C: these data should be backed up by a phosphorylated TBK1 western blots.

These have been added and quantified from three independent replicates (Fig. EV3 D)

16. Where possible addition of more healthy donors for comparison would improve the robustness of this study (eg: Figure 2E-F; Figure 4C, etc) as basal ISG levels (etc) will vary between individuals. I commend the comparison of 29 healthy donors in Figure 1A and hope the authors can extend this to include additional donors in fibroblast experiments.

As the reviewer suggests, we included more healthy donor fibroblasts even in technically challenging settings and re-performed selected key experiments to include 2 more additional donors (new Fig. 2 E, F; new Fig. 5D, E; new Fig. EV4 B, C).

Text:

Overall the text is fine, there are a few sentences that don't read well for me (eg: line 50: disrupt should be disrupts, line 122: starting a new paragraph with a joining phrase "in line" is confusing) but this should all be picked up in the editing process. I have picked up a couple phrases which need to be rewritten for scientific clarity:

Line 147: "In line, in 293-Dual hSTING-R232 cells, which endogenously express human STING" should be rewritten, these do not endogenously express STING. Exogenous STING has been stably expressed in this line.

Line 199: "To terminate DNA sensing, STING is transported in an ARF1-dependent manner from the Golgi to the ER." I understand the gist of this sentence, but it should be rewritten, STING does not directly sense DNA and STING signalling can be halted by transport to lysosomes for degradation.

These textual changes were implemented.

More details in figure legends would improve readability of this manuscript.

More detail has been added.

Overall, I commend the authors on their work and wish them luck with their follow up experiments

Thank you, we appreciate the detailed and constructive feedback.

Referee #3:

This manuscript by Lang et al is a follow up to their work published last year in Nature Communications (PMID 37914730), which described a disease-causing mutation (R99) in the GTPase ARF1, thereby driving type I IFN-driven pathology. This current manuscript follows the same template in characterising a separate mutation (R19C), performing many of the same assays which were used in the previous paper. While the novelty is limited given the similarities to their previous work, this is still an elegant, well-conducted piece of work which has also shed further light on the dual role of ARF1 in regulating mitochondrial dynamics and termination of STING signalling. I have some minor suggestions as to how the manuscript might be improved, which are listed below:

Thank you very much for the positive comments.

1. In line 213, the authors claim to have provided evidence that 'active' STING accumulates at the ERGIC/GOLGI. However, active STING would indicate that they have measured levels of phosphorylated STING, which they have not. The authors could measure p-STING in the immunofluorescence experiments they have performed or simply measure it by Western Blot in their mutant fibroblasts.

This reviewer is correct, we associated p-TBK1 presence with activation of STING. We have toned down our results section accordingly (lines 215-216).

2. The immunoprecipitation experiments (Fig. 3B) are important in demonstrating a reduced ARF1-COPI interaction in mutant cells. However, the authors performed this experiment in an overexpression system which might be considered slightly artificial. Given the importance of this experiment, would it be possible to perform the immunoprecipitation endogenously in patient fibroblasts?

The immunoprecipitation experiments have been repeated in primary fibroblasts (healthy control, R19C patient) (new EV2 D), essentially showing the same results as the overexpression experiment.

3. Fig. 2D is the only figure which I have a slight conceptual issue with. The data derived from patient samples (Fig. 1A and Figs 5F and G) show the presence of an interferon response in R19C even in the absence of stimulation, yet experiments in the hSTING-R232 cells indicate that STING must be active in order to obtain an interferon response in the R19C mutant. I assume that the difference owes to the primary human cells already exhibiting a basal level of active STING. However, some readers might find these results slightly contradictory, so it may be worth including a line of discussion concerning this point.

We agree and have amended our discussion (line 305-315).

4. In lines 129-132, the authors compare the doses of the respective mutant plasmids which are required to induce IFN signalling. I think that it is tricky to make this claim given that increased plasmid dose does not necessarily equal increased expression as expression patterns may differ between plasmids. I would suggest the authors remove this line.

We agree with the reviewer and have removed the exact values for the doses and the quantitative IC50. However, we note that expression levels as shown on western blot (updated Fig. 1D) are comparable (at different doses) between ARF1 R19C and ARF R99C.

5. It would be great if the authors could include electron microscopy images of the R99C mutant alongside the WT and R19C mutant in Figure 2A for comparison

New electron microscopy images of the R99C mutant have been added (updated Fig. 2A).

6. Along the same lines, could the authors move the R99C colocalization data (Fig. EV3B) alongside the R19C colocalization data in Fig. 3D for comparison?

The vector and ARF1 R99C colocalization data have been moved next to the R19C colocalization data (updated Fig. 3E).

7. Fig. 5D shows an increased IFN response in the mutant patient fibroblasts when stimulated with cGAMP. As an additional control to demonstrate that cGAS-STING signalling specifically is heightened in these cells, could the authors also stimulate the cells with a different PRR agonist (e.g. LPS)? Theoretically, LPS-induced responses should be similar between wild-type and mutants

We have performed the experiment as suggested, with the analysis revealing that patient fibroblasts are not hyper responsive to Poly I:C or LPS stimulation (new Fig. EV4 B,C and EV5B).

Dear Dr. Sparrer

Thank you for the submission of your revised manuscript to EMBO reports. We have now received the reports from the referees who were asked to assess it (copied below).

As you will see, both referees are positive and consider the revised manuscript strengthened. That said, referee #2 raises several points that need to be addressed. I also noticed the reduced expression of gamma-COP in ARF1 R19C fibroblasts compared to WT. The reduced expression might explain why ARF1 binding to gamma-COP is not detectable. Please explain and discuss these results and tone down related conclusions appropriately. Please also provide data on autophagic flux in mutant fibroblasts.

From the editorial side, there are also a few things that we need before we can proceed with the official acceptance of your study.

- Your manuscript will be published in our Reports section, we would therefore need a combined Results and Discussion section.
- Please move the Data Availability paragraph to the end of the Methods section, before the Acknowledgements.
- Regarding the Author Contributions, we now use CRediT to specify the contributions of each author in the journal submission system. Therefore, please remove the Author Contributions from the manuscript file and make sure that the author contributions in our online manuscript tracking system are correct and up-to-date. The information you specified in the system will be automatically retrieved and typeset into the article. You can enter additional information in the free text box provided, if you wish.
- Please provide an ORCID for Dr. Hirschenberger. This information is mandatory for all corresponding authors.
- Please upload the file called "Supplementary Patient Description" as PDF called "Appendix", file type "Supplementary Information". Please change the callout to this file in line 610 to "Appendix" instead of "Supplementary Information file".
- During our routine image integrity checks we noticed that the FLAG-ARF1 blots from Figure 1 C & D have been reused in Figure EV1 A & B. The same is true for Figure 2D and Figure EV1 E. Please indicate this reuse in the respective figure legends.
- The FLAG blots in Figure 2B seem to have high contrast settings. If possible, i.e., if the original blots show lower contrast/brightness, please reduce these parameters in the figure.
- In figure EV2A, the vector control FLAG image seems without signal or noise. I understand that this is the negative control, which does not express FLAG but in order to avoid any ambiguity, could you please provide the source data for this panel and also for Figure EV3A? Thank you very much.
- Figure 5A, EV4A: please define the scale bar size only in the legend and remove the numbers from the scale bar in the images.
- Please provide the exact p values in the legends of figures 1B, C; 2C-F; 3D, E; 4B, D, F; 5B-H; EV1 C, D; EV2 B; EV3 C, D; EV5 D.
- As a standard procedure, we edit the title and abstract of manuscripts to make them more accessible to a general readership. Please find my suggestion below my signature.

With kind regards,

=====

Referee #2:

I commend the authors on their excellent research

Referee #3:

Fig. EV2D. The authors have performed an endogenous immunoprecipitation experiment to examine the ARF1-COPI interaction in primary fibroblasts, as I requested in my initial reviewer report. The authors acknowledge this in the text but total WCL γ -COP levels were lower in ARF1 R19C fibroblasts compared to WT fibroblasts. As a result the authors cannot claim that ARF1 to γ -COP binding was 'undetectable' in the mutant fibroblasts as there is barely any γ -COP there in the first place. The description of these results must therefore be rewritten to avoid misinterpretation. Can the authors provide any explanation for this reduction in basal γ -COP levels in the patient fibroblasts?

I see that the authors have provided new data examining autophagy in patient fibroblasts (Fig. EV4D). As I understand it, simply measuring LC3B-II expression is not an appropriate way to determine autophagic activity. The authors must measure autophagic flux in the presence of an autophagy inhibitor (e.g. bafilomycin) in order to properly address the question of whether autophagy is altered in mutant fibroblasts

In lines 246-247, the authors provide the extent of STING knockout by western blot as a percentage. I do not think this is necessary/possible given the qualitative nature of western blot- it is sufficient to simply say that STING was depleted in both control and patient fibroblasts

=====

Abstract

The cGAS-STING cascade detects cytosolic DNA to induce type I interferon (IFN) responses but tight control of DNA sensing is crucial to avoid auto-inflammation. The GTPase ADP-ribosylation factor 1 (ARF1) plays a major role in maintaining cGAS-STING homeostasis and various pathogenic ARF1 variants are associated with type I interferonopathies. Functional ARF1 plays a dual role in inhibiting STING activity. It maintains mitochondrial integrity preventing mtDNA release and it facilitates COPI-mediated retrograde STING trafficking and deactivation, yet the factors governing these distinct functions of ARF1 remained unexplored. Here, we dissect ARF1's dual role by a comparative analysis of disease-associated ARF1 variants and their impact on STING signaling. We identify a de novo heterozygous s.55C>T/p.R19C ARF1 variant in a patient with symptoms of a type I interferonopathy and elevated STING signaling. The GTPase-deficient variant ARF1 R19C selectively disrupts retrograde transport of STING and COPI binding without affecting mitochondrial integrity, prolonging innate immune activation. Treatment of patient fibroblasts in vitro with the STING signalling inhibitors H-151 and amlexanox reduces chronic IFN signalling. Summarizing, our data reveal the molecular basis of a novel ARF1-associated type I interferonopathy that allows dissection of the two roles of ARF1, suggesting that pharmacological targeting of STING may alleviate ARF1-associated auto-inflammation.

Point-by-Point response to referees and editorial comments EMBOR-2024-60207V2 Answers highlighted in blue

Referee #3:

Fig. EV2D. The authors have performed an endogenous immunoprecipitation experiment to examine the ARF1-COPI interaction in primary fibroblasts, as I requested in my initial reviewer report. The authors acknowledge this in the text but total WCL γ -COP levels were lower in ARF1 R19C fibroblasts compared to WT fibroblasts. As a result the authors cannot claim that ARF1 to γ -COP binding was 'undetectable' in the mutant fibroblasts as there is barely any γ -COP there in the first place. The description of these results must therefore be rewritten to avoid misinterpretation. Can the authors provide any explanation for this reduction in basal γ -COP levels in the patient fibroblasts?

We agree and have now rephrased this part/toned down the conclusion: "Endogenous ARF1 readily co-precipitated γ -COP in healthy donor fibroblasts (Fig. EV2D). A γ -COP-ARF1 interaction in primary ARF1 R19C patient fibroblasts could not be detected, however this was potentially occluded by very low basal γ -COP levels in the patient fibroblasts." (line 180-183). Currently, we cannot provide a precise explanation of this phenotype, it, however, may be related to the general COPI trafficking disturbance induced by patient ARF1 R19C. Of note, temporary overexpression of ARF1 R19C does not induce γ -COP degradation in HEK293T cells (Fig. 3C). We have included a short discussion in line 215-216.

I see that the authors have provided new data examining autophagy in patient fibroblasts (Fig. EV4D). As I understand it, simply measuring LC3B-II expression is not an appropriate way to determine autophagic activity. The authors must measure autophagic flux in the presence of an autophagy inhibitor (e.g. bafilomycin) in order to properly address the question of whether autophagy is altered in mutant fibroblasts

The experiment was performed as the reviewer suggested. Quantification of three independent repeats revealed that autophagic flux is not altered in ARF1 R19C patient cells compared to healthy donor cells (new Fig. EV4C). This is in line with our previous data on STING degradation kinetics (Fig. EV4D).

In lines 246-247, the authors provide the extent of STING knockout by western blot as a percentage. I do not think this is necessary/possible given the qualitative nature of western blot- it is sufficient to simply say that STING was depleted in both control and patient fibroblasts

This section was updated. "Western blot analysis showed that endogenous levels of STING were strongly reduced in healthy control fibroblasts and patient AGS3238 (R19C) fibroblasts by Cas9/gRNA treatment (Fig. 5F)." (line 295-297).

Editorial Comments:

- Your manuscript will be published in our Reports section, we would therefore need a combined Results and Discussion section.

We have merged the discussion into the results section (major changes highlighted in yellow) without compromising the content of the reviewed discussion.

- Please move the Data Availability paragraph to the end of the Methods section, before the Acknowledgements.

This was done.

- Regarding the Author Contributions, we now use CRediT to specify the contributions of each author in the journal submission system. Therefore, please remove the Author Contributions from the manuscript file and make sure that the author contributions in our online manuscript tracking system are correct and up-to-date. The information you specified in the system will be automatically retrieved and typeset into the article. You can enter additional information in the free text box provided, if you wish.

Done.

- Please provide an ORCID for Dr. Hirschenberger. This information is mandatory for all corresponding authors.

This was submitted into the system. For reference, the ORCID of Dr. Hirschenberger is 0000-0001-8766-1271.

- Please upload the file called "Supplementary Patient Description" as PDF called "Appendix", file type "Supplementary Information". Please change the callout to this file in line 610 to "Appendix" instead of "Supplementary Information file".

Done.

- During our routine image integrity checks we noticed that the FLAG-ARF1 blots from Figure 1 C & D have been reused in Figure EV1 A & B. The same is true for Figure 2D and Figure EV1 E. Please indicate this reuse in the respective figure legends.

We have removed the panels that were shown twice.

- The FLAG blots in Figure 2B seem to have high contrast settings. If possible, i.e., if the original blots show lower contrast/brightness, please reduce these parameters in the figure.

Done. The respective Source data has been updated too.

- In figure EV2A, the vector control FLAG image seems without signal or noise. I understand that this is the negative control, which does not express FLAG but in order to avoid any ambiguity, could you please provide the source data for this panel and also for Figure EV3A? Thank you very much.

We have reinserted the panel with a lower contrast cutoff setting, the background/noise is now visible. The raw data has been included in the Source data.

- Figure 5A, EV4A: please define the scale bar size only in the legend and remove the numbers from the scale bar in the images.

Done. Has also been done for Figs. 4C, E, EV 3A, B.

- Please provide the exact p values in the legends of figures 1B, C; 2C-F; 3D, E; 4B, D, F; 5B-H; EV1 C, D; EV2 B; EV3 C, D; EV5 D.

Done.

- As a standard procedure, we edit the title and abstract of manuscripts to make them more accessible to a general readership. Please find my suggestion below my signature.

The altered abstract was included in the manuscript, albeit it had to be shortened to 175 words for submission. We would appreciate it if you double-check whether this shortened version is still accessible enough.

Dr. Konstantin Sparrer
German Center for Neurodegenerative Diseases
Neurovirology and Neuroinflammation
Meyerhofstr.1
Ulm, Baden-Württemberg 89081
Germany

Dear Dr. Sparrer,

I am very pleased to accept your manuscript for publication in the next available issue of EMBO reports. Thank you for your contribution to our journal.

Yours sincerely,
